## 1 Toward a typology of river functioning: a

## 2 comprehensive study of the particulate organic

## 3 matter composition at the multi-river scale

- Florian Ferchiche<sup>1</sup>, Camilla Liénart<sup>1</sup>, Karine Charlier<sup>1</sup>, Jonathan Deborde<sup>2,3</sup>, Mélanie Giraud<sup>4</sup>, Philippe
- Kerhervé<sup>5</sup>, Pierre Polsenaere<sup>1,3</sup>, Nicolas Savoye<sup>1\*</sup>
- <sup>1</sup> Univ. Bordeaux, CNRS, EPHE, Bordeaux INP, UMR 5805 EPOC, F-33600 Pessac, France
- Univ. Pau & Pays Adour, CNRS, E2S UPPA MIRA, UMR 5254 IPREM, F-64000 Pau, F-64600 Anglet, France
- <sup>3</sup> Ifremer, COAST, F-17390 La Tremblade, France
- <sup>4</sup> MNHN, Station Marine de Dinard, F-35800 Dinard, France
- <sup>5</sup> Univ. Perpignan, CNRS, UMR 5110 CEFREM, F-66860 Perpignan, France

- \*Correspondence to: Nicolas Savoye (<u>nicolas.savoye@u-bordeaux.fr)</u> (+33)556223916
- Université de Bordeaux UMR 5805 EPOC Station marine d'Arcachon 2 rue du Professeur Jolyet 33120
- Arcachon
- Keywords: river-estuary interface; particulate organic matter; stable isotopes; multi-ecosystem
- study.

#### **Abstract**

44

19 In riverine systems, particulate organic matter (POM) originates from various sources, each 20 having its proper dynamics related to production, decomposition, transport and burial, resulting 21 in a significant spatiotemporal heterogeneity in the POM pool. The current study, based on C 22 and N elemental and isotopic ratios, applies Bayesian mixing models associated with statistical 23 multivariate analyses to 1) quantify and examine relationships between POM composition and 24 environmental forcings, and 2) draw a typology of river functioning based on POM composition 25 and its seasonal dynamics. 26 Twenty-three temperate rivers accounting for a large diversity of environmental conditions 27 were sampled fortnightly to monthly for one to seven years at their River-Estuary Interface 28 (REI). Phytoplankton and labile terrestrial material occurred in all rivers, whereas sewage and 29 refractory terrestrial material were present in only a few. At the twenty-three rivers scale, 30 phytoplankton dominance was associated with agricultural surfaces, while labile terrestrial 31 material was linked to organic-rich leached soil and refractory terrestrial matter to steep 32 catchments with little soil. Seasonal dynamics were primarily driven by phytoplankton growth, 33 river discharge (labile terrestrial material), and sediment resuspension (refractory terrestrial 34 material). 35 A statistical regionalisation defined four types of river dynamics: (1) systems whose POM is 36 dominated by labile terrestrial material year-round; (2) systems whose POM is composed of 37 labile and refractory terrestrial material, in addition to phytoplankton, showing variable 38 seasonality; systems whose POM is composed of phytoplankton and labile terrestrial material 39 (3) without and (4) with pronounced seasonality. This work offers a comprehensive understanding of POM composition, spatio-temporal dynamics and controlling factors at the 40 41 REI in temperate climates, complementing similar work dedicated to coastal systems. Future 42 work dedicated to estuaries is called to get a comprehensive understanding of POM 43 composition, dynamics and drivers along the Land-Ocean Aquatic Continuum.

#### 1. Introduction

45

The River-Estuary Interface (REI) is a crucial biogeochemical interface for understanding the 46 47 transition between continental and coastal systems, beginning at estuaries, because of its key 48 location within the Land-Ocean Aquatic Continuum (LOAC) (Bate et al., 2002). Indeed, rivers 49 then estuaries are important filters for matters received from land, transporting and transforming 50 organic matter and nutrients along their courses (Bouwman et al., 2013; Dürr et al., 2011; 51 Middelburg and Herman, 2007). These processes are fundamental in understanding global 52 biogeochemical cycles (Regnier et al., 2013), as these matters directly fuel coastal ocean trophic 53 networks (Dagg et al., 2004). However, in a Human-impacted world, anthropogenic activities 54 and disturbances can modify natural matter fluxes. For example, damming rivers directly 55 impacts nutrient flows (Wang et al., 2022) and sediment transportation (Kang et al., 2021). Indirectly, land use in river basins can lead to changes in the river matter quality (Lambert et 56 57 al., 2017). 58 In aquatic systems, particulate organic matter (POM), i.e., non-mineral particles, is composed 59 of different sources that originate from different compartments: phytoplankton, macrophytes 60 from the aquatic systems, as well as plant litter, soil and petrogenic particles from terrestrial 61 compartments and even treated and untreated anthropogenic organic matter (Ke et al., 2019; 62 Sun et al., 2021; Zhang et al., 2021). Depending on its composition, POM exhibits different 63 levels of lability, i.e., different levels of biogeochemical reactivity and bioavailability. For 64 instance, phytoplankton is usually considered highly labile, and thus highly biogeochemically 65 reactive and bioavailable for primary consumers, while terrestrial POM is usually considered 66 more and more refractory through degradation processes, thus lightly biogeochemically 67 reactive and poorly bioavailable for the food webs (Brett et al., 2017; David et al., 2005; Etcheber et al., 2007). In other words, the determination and quantification of POM 68 69 composition (i.e., the relative proportion of each source composing the POM) allow a better 70 understanding of biogeochemical cycles and trophic ecology in aquatic systems (e.g., Grunicke 71 et al., 2023; Minaudo et al., 2016). Nevertheless, POM composition and concentration are not 72 only involved in biogeochemical and biological processes (e.g., primary production, 73 remineralisation, feeding) but also undergo other processes inside and at the interface of the 74 aquatic compartment (Canuel and Hardison, 2016). River hydrology is a key factor controlling 75 POM composition and concentration. The bedrock and soil characteristics of each catchment, 76 together with climatic conditions, shape the erosional processes, leading to great variabilities in 77 hydrodynamics, terrestrial material quality and quantity and phytoplankton growth conditions 78 (Dalzell et al., 2007; Hilton et al., 2010; Lebreton et al., 2016). This variability leads to shifts 79 in POM source origins (Arellano et al., 2019; Barros et al., 2010). Additionally, anthropogenic 80 disturbances can directly or indirectly affect seasonal as well as long-term patterns of POM 81 composition and concentration. For instance, a decrease in nutrient load affects phytoplankton 82 production and biomass (Minaudo et al., 2015), agricultural surface alters soil properties and

- erosion and consequently soil particle export to rivers, damming alters river hydromorphology
- and consequently particle dynamics and export (Kang et al., 2021; Zhang et al., 2021), etc.
- This dependency of POM composition and concentration to physical, biogeochemical and
- biological processes and their responses to environmental conditions and characteristics (Bonin
- et al., 2019; Falkowski et al., 1998; Field et al., 1998; Galeron et al., 2017; Goñi et al., 2009;
- Lebreton et al., 2016) may lead to distinguishing different types of rivers, i.e., the likeliness of
- rivers to carry preferential sources. For instance, highly turbid systems are more likely to carry
- refractory materials (Savoye et al., 2012), while eutrophicated rivers carry high biomass of
- phytoplankton (Hounshell et al., 2022; Minaudo et al., 2015), and contrasting processes can
- lead to a mixture between different detrital sources, as soil matter vs. fresh terrestrial plants
- (Ogrinc et al., 2008). However, to date, no study clearly determined typologies of river
- dynamics based on POM composition and its seasonal variability.
- To distinguish POM sources and quantify their contribution to POM composition, different
- tools such as elemental and isotopic ratios, pigments or specific compounds like fatty acids or
- alkanes can be used (e.g., Chevalier et al., 2015; Liénart et al., 2020, 2017; Savoye et al., 2012).
- Elemental and isotopic ratios are usually considered robust and allow the quantification of POM
- composition in this kind of study (e.g., Liénart et al., 2016; Onstad et al., 2000; Wang et al.,
- 2021). Indeed, they usually allow the discrimination of, e.g., riverine phytoplankton, terrestrial
- POM and wastewater POM (Ke et al., 2019), and they can be used for running mixing models
- that quantify the proportion of the different sources in a POM mixture (Parnell et al., 2013).
- However, studies using mixing models for quantifying POM composition in river systems are
- still quite scarce (e.g., Ferchiche et al., 2025, 2024; Kelso and Baker, 2022, 2020; Zhang et al.,
- 2021).

115

- Within the scope of better understanding the role of the LOAC in modifying matter fluxes and
- quality, the present study gathered published data and results from 23 rivers at the river-estuary
- interface with the aim of 1) quantifying the POM composition of each river, 2) describing the
- seasonal variations of this composition, 3) determining the drivers of the seasonal variability
- within each river and the spatial variability among the 23 rivers, and then 4) determining a
- typology of river dynamics according to their POM composition and dynamics. This study is
- the first that precisely quantify POM composition in numerous and various temperate river
- systems in a world region (here, Western Europe) and classify river types according to POM
- composition and dynamics.

#### 2. Materials and methods

- Twenty-three temperate rivers were studied at their river-estuary interface (i.e., right upstream
- of the tidal influence). All the data come from published studies or national open databases. To
- minimize the heterogeneity of the datasets in terms of sampling strategy, we have considered
- for this study the datasets only when 1) C/N ratio along with isotopic ratio of carbon and/or

nitrogen were available, 2) particulate matter characteristics like, suspended particulate matter (SPM), particulate organic carbon (POC), particulate nitrogen (PN) or chlorophyll *a* (chl *a*) were also available, 3) datasets exhibited at least a monthly temporal resolution for one full year. When needed, published datasets were completed and harmonised thanks to national databases.

#### 2.1. Study sites

The studied rivers and associated watersheds are all located in France (except the upper basin of the Rhône River) and distributed in all regions of the mainland. Three, fifteen and five of these rivers flow into the English Channel, the Atlantic Ocean and the Mediterranean Sea (Fig. 1). Even if located in a somewhat restricted area (Western Europe) at the global scale, they encompass large gradients of environmental characteristics (Tab. 1) for a temperate climate. For instance, the Loire River is one of the largest in Europe (length: 1006 km; watershed:  $117,356 \text{ km}^2$ ), while the littlest studied river is a very small stream of the Arcachon lagoon (length: 3 km; watershed:  $18 \text{ km}^2$ ). They encompass large gradients of river flow (annual mean:  $0.3 \text{ m}^3/\text{s} - 1572 \text{ m}^3/\text{s}$ ), turbidity (SPM annual mean: 2.7 mg/l - 40.9 mg/l) and trophic status (from oligotrophic to eutrophic rivers; chlorophyll *a* annual mean: 0.4 µg/l - 57.1 µg/l). At last, they undergo a gradient of anthropic pressures as illustrated by the proportion of artificial surfaces (0.1 % - 5.6 %) and agricultural areas (0 % - 86 %) in the watersheds (Fig. 1).

<u>Figure 1</u> Studied rivers (thick blue lines), sampling locations (black stars) and watersheds (thin black lines), including the main land uses (red, yellow and green colours). 1: Seine; 2: Orne; 3: Rance; 4: Elorn; 5: Aulne; 6: Loire; 7: Sèvre niortaise; 8: Charente; 9: Seudre; 10: Canal du Porge; 11: Cirès; 12: Milieu; 13: Lanton; 14: Renet; 15: Tagon; 16: Leyre; 17: Canal des Landes; 18: Adour; 19: Têt; 20: Aude; 21: Orb; 22: Hérault; 23: Rhône. Background: ESRI Ocean

suspended particulate matter (SPM) and chlorophyll a (chl a). Id: identification number; Number: number of sampling dates. River types were defined within the scope of the present study (see section 3.4). Table 1 Overview of river samplings and characteristics. Values are given as annual mean over the study period for river flow, temperature,

| Rhône                                                                              | Aude                  | Orb                   | Hérault               | Têt                   | Adour                                   | Landes                  | Leyre                                                                            | Tagon                   | Milieu                  | Lanton                  | Renet                   | Cirès                   | Porge                   | Seudre                     | Charente                   | Sèvre                      | Loire                  | Aulne                      | Elorn                      | Rance                      | Orne                       | Seine                      | River                           |
|------------------------------------------------------------------------------------|-----------------------|-----------------------|-----------------------|-----------------------|-----------------------------------------|-------------------------|----------------------------------------------------------------------------------|-------------------------|-------------------------|-------------------------|-------------------------|-------------------------|-------------------------|----------------------------|----------------------------|----------------------------|------------------------|----------------------------|----------------------------|----------------------------|----------------------------|----------------------------|---------------------------------|
| 23                                                                                 | 22                    | 21                    | 20                    | 19                    | 18                                      | 17                      | 16                                                                               | 15                      | 14                      | 13                      | 12                      | 11                      | 10                      | 9                          | ∞                          | 7                          | 6                      | 5                          | 4                          | သ                          | 2                          | 1                          | Id                              |
| П                                                                                  | П                     | П                     | п                     | П                     | Ħ                                       | H                       | н                                                                                | П                       | Н                       | П                       | Н                       | П                       | Ħ                       | П                          | Ħ                          | V                          | V                      | V                          | V                          | V                          | Ħ                          | $\mathbf{N}$               | River<br>type                   |
| 12/2003 to 01/2011                                                                 | 01/2006 to 05/2010    | 01/2006 to 05/2010    | 01/2006 to 05/2010    | 01/2006 to 05/2010    | 04/2013 to 06/2014<br>and 05/2017 to    | 02/2008 to 02/2009      | 01/2008 to 03/2010<br>and 02/2014 to<br>02/2015                                  | 02/2008 to 02/2009      | 01/2008 to 02/2009      | 03/2014 to 09/2015         | 03/2014 to 03/2015         | 03/2014 to 03/2015         | 10/2009 to 07/2012     | 01/2014 to 06/2015         | 01/2014 to 06/2015         | 06/2014 to 05/2015         | 06/2014 to 06/2015         | 06/2014 to 06/2015         | Sampled period                  |
| monthly                                                                            | monthly               | monthly               | monthly               | monthly               | monthly                                 | monthly                 | bi-monthly or<br>monthly                                                         | bi-monthly              | monthly                 | monthly                 | bi-monthly              | monthly                 | monthly                 | monthly                    | monthly                    | monthly                    | bi-monthly             | monthly                    | monthly                    | monthly                    | monthly                    | monthly                    | Sampling<br>Periodicity         |
| 105                                                                                | 52                    | 52                    | 52                    | 52                    | 24                                      | 12                      | 59                                                                               | 26                      | 13                      | 13                      | 23                      | 13                      | 14                      | 15                         | 13                         | 13                         | 67                     | 17                         | 17                         | 12                         | 13                         | 13                         | Num<br>ber                      |
| 43.6787                                                                            | 43.2442               | 43.2850               | 43.3594               | 42.7137               | 43.4988                                 | 44.6169                 | 44.6263                                                                          | 44.6590                 | 44.6973                 | 44.7002                 | 44.7144                 | 44.7598                 | 44.7898                 | 45.6740                    | 45.8680                    | 46.3153                    | 47.3920                | 48.2127                    | 48.4505                    | 48.4916                    | 49.1797                    | 49.3067                    | Latitud<br>e                    |
| 4.6212                                                                             | 3.1527                | 3.2813                | 3.4354                | 2.9935                | -1.2949                                 | -1.1091                 | -0.9961                                                                          | -0.9891                 | -1.0225                 | -1.0244                 | -1.0441                 | -1.1107                 | -1.1612                 | -0.9331                    | -0.7131                    | -1.0039                    | -0.8604                | -4.0944                    | -4.2483                    | -2.0014                    | -0.3491                    | 1.2425                     | Num Latitud Longitu<br>ber e de |
| 812                                                                                | 223                   | 136                   | 148                   | 115                   | 308                                     | 14                      | 116                                                                              | 10                      | 7                       | 15                      | 3                       | 12                      | 57                      | 68                         | 381                        | 158                        | 1006                   | 144                        | 56                         | 103                        | 169                        | 774                        | River length (km)               |
| 95590                                                                              | 5327                  | 1585                  | 2582                  | 1369                  | 16912                                   | 117                     | 1700                                                                             | 30                      | 21                      | 36                      | 18                      | 45                      | 222                     | 855                        | 9855                       | 3650                       | 117356                 | 1875                       | 385                        | 1195                       | 2932                       | 79000                      | Catchment<br>area (km²)         |
| 1572                                                                               | 40                    | 23                    | 53                    | 23                    | 516                                     | 0.49                    | 17                                                                               | 0.64                    | 0.58                    | 0.26                    | 0.56                    | 0.58                    | 3.48                    | 1.81                       | 68                         | 3.72                       | 630                    | 30                         | 6                          | 1.37                       | 16                         | 496                        | River flow (m3/s)               |
| 15.9                                                                               | 14.2                  | 15.7                  | 16.0                  | 15.7                  | 14.0                                    | 14.1                    | 13.0                                                                             | 12.6                    | 12.7                    | 12.5                    | 12.9                    | 12.2                    | 13.3                    | 14.3                       | 15.1                       | 15.7                       | 14.1                   | 14.4                       | 12.3                       | 15.1                       | 14.5                       | 15.0                       | Water<br>temperat<br>ure (°c)   |
| 41                                                                                 | 31                    | ∞                     | 7                     | <b>%</b>              | 48                                      | 3                       | 11                                                                               | 13                      | 7                       | 1                       | 10                      | 5                       | 12                      | 17                         | 13                         | 13                         | 19                     | 7                          | 16                         | 21                         | =                          | 21                         | SPM<br>(mg/l)                   |
| 1.9                                                                                | NA                    | NA                    | NA                    | NA                    | 2.4                                     | 1.1                     | 0.9                                                                              | 1.3                     | 0.4                     | 1.2                     | 0.6                     | 0.4                     | 5.0                     | 0.5                        | 1.3                        | 3.8                        | 18.7                   | 3.3                        | 3.0                        | 57.1                       | 1.8                        | 2.8                        | Chl <i>a</i> (µg/l)             |
| Harmelin-Vivien et al., 2010<br>/ Cathalot et al., 2013 /<br>Higueras et al., 2014 | Higueras et al., 2014 | Liénart et al., 2016 /<br>Deborde, 2019 | Polsenaere et al., 2013 | Dubois et al., 2012 /<br>Polsenaere et al., 2013 /<br>Liénart et al., 2017, 2018 | Polsenaere et al., 2013 | Liénart et al., 2017, 2018 | Liénart et al., 2017, 2018 | Liénart et al., 2017, 2018 | Ferchiche et al., 2024 | Liénart et al., 2017, 2018 | References                      |

#### 2.2.Data origin

Regarding the core parameters (C/N ratio,  $\delta^{13}$ C,  $\delta^{15}$ N, water temperature, SPM, POC, PN, chl 148 149 a), most of the data sets come from published studies (Canton et al., 2012; Cathalot et al., 2013; 150 Dubois et al., 2012; Ferchiche et al., 2024; Harmelin-Vivien et al., 2010; Higueras et al., 2014; 151 Liénart et al., 2016, 2017, 2018; Polsenaere et al., 2013), while most of additional parameters 152 come from national databases (Tab. A1). When not available in the cited studies, concentrations 153 of SPM, NO<sub>3</sub>-, NH<sub>4</sub><sup>+</sup> and PO<sub>4</sub><sup>3</sup>-, pH and water temperature were retrieved from the *Naïades* 154 database (https://naiades.eaufrance.fr/, consulted the 07/10/2023). Note that these parameters 155 were not necessarily measured or sampled exactly at the same location or date for Naïades as 156 in the cited studies. In that case, the location was chosen as close as possible to the study 157 location and data values were time-interpolated to match the study date. Meteorological 158 variables (air temperature, zonal and meridional wind, irradiance; used to qualify 159 photosynthetic favourable conditions or wind-induced resuspension) come from Météo France, 160 the French meteorological service. Wind data were received originally as direction and speed. 161 To remove the angular bias, they were combined using scalar products to get zonal and 162 meridional wind speeds, which range between minus and plus infinity (see Lheureux et al., 2022, for more details). River flows (used to qualify the hydrodynamics forcing) were retrieved 163 164 from the *Banque Hydro* database (https://www.hydro.eaufrance.fr/, consulted the 07/10/2023) 165 or from Polsenaere et al. (2013) for the small streams. 166 Catchment properties were retrieved when available for the 23 rivers. Land use proportions 167 originate from the Corine Land Cover database (https://www.statistiques.developpement-168 durable.gouv.fr/corine-land-cover-0, consulted the 10/01/2024). Soil organic carbon data 169 originate from the SoilTrEC database (https://esdac.jrc.ec.europa.eu/content/predicted-170 distribution-soc-content-europe-based-lucas-biosoil-and-czo-context-eu-funded-1, 171 the 10/01/2024). Net erosion soil data originate from the WaTEM/SEDEM database 172 (https://esdac.jrc.ec.europa.eu/content/estimate-net-erosion-and-sediment-transport-usingwatemsedem-european-union, consulted the 10/01/2024). Strahler numbers originate from the 173 174 CARTHAGE database (https://www.sandre.eaufrance.fr/atlas/srv/api/records/c1d89cc3-c530-4b0d-b0ae-06f5ebf7997d, consulted the 15/08/2025). Useful water reserve values come from 175 176 the GSF database (Le Bas, 2025). Bedrock, soil types, granulometry and slope come from the 177 GISSOL database (gathered by great bedrock and soil types according to the provided legend; 178 INRA, 2025). Wastewater treatment capacities originate from the Eau France WFS services 179 (https://services.sandre.eaufrance.fr/geo/odp?REQUEST=getCapabilities&service=WFS&VE 180 RSION=2.0.0, couche sa:SysTraitementEauxUsees, consulted the 15/08/2025). All these 181 catchment data were pre-processed on a Geographical Information System to extract 182 information for each catchment surface, then averaged or weighted (depending on continuous 183 or semi-quantitative data) to characterise each system with a value.

It should be noted that a complete study was already dedicated to the Loire River and reported 185 as a companion article (Ferchiche et al., 2024). Consequently, the results are not reported in the present study but are used for multi-system comparisons (Fig. 5 and 7, and corresponding text). 186

#### 2.3. Determination of sources signatures

208

- To run mixing models for quantifying POM composition, it is previously needed to 1) 189 determine sources of POM, and 2) associate elemental and isotopic signatures to these sources. 190 In riverine systems, autochthonous (mainly phytoplankton) and allochthonous (resuspended 191 sediment, terrestrial fresh litter or rock-derived soil) are the main sources that are usually 192 considered as fuelling the POM (e.g., Ferchiche et al., 2024; Pradhan et al., 2016; Sarma et al., 193 2014). Nevertheless, sewage POM may also contribute (Higueras et al., 2014). Consequently, 194 phytoplankton, labile and refractory terrestrial POM and sewage POM were considered as 195 potential sources in this study.
  - Phytoplankton cannot be easily picked up from bulk particles to measure its elemental and isotopic ratios. Therefore, the method developed and used by Savoye et al. (2012), Liénart et al. (2017) and Ferchiche et al. (2024) was applied here. It consists of determining the elemental and isotopic ratios from a subset of the bulk dataset. Briefly, phytoplankton-dominated POM is characterised by a low POC/chl a ratio ( $\leq 200$  or even  $\leq 100$  g/g; Savoye et al., 2003 and references therein). Thus, elemental and isotopic ratios of samples exhibiting a low POC/chl a ratio can be considered as good estimates of phytoplankton elemental and isotopic ratios. When the POC/chl a ratio is not available, samples exhibiting a high PN/SPM ratio can be used. Additional constraints may be used to minimise potential overlap between phytoplankton and terrestrial elemental and isotopic signatures. Phytoplankton elemental and especially isotopic ratios may vary deeply over time and space depending on primary production intensity and potential limiting factors, nutrient origin, etc. (e.g., Miller et al., 2013; Savoye et al., 2003). When existing, this variability has to be taken into account to avoid using elemental and isotopic signatures that are not valid at the time or location of the sampling. This could be performed by using regressions between elemental and/or isotopic ratios and environmental variables (see Ferchiche et al., 2024; Liénart et al., 2017; Savoye et al., 2012). At last, when no samples exhibit a low POC/chl a ratio, samples exhibiting the lowest (even if high) POC/chl a ratios can be used, but the data should first be corrected for the contribution of the terrestrial POM using Equations 1-3.

$$\delta^{13}C_{sample} = \left([POC]_{phytoplankton} \times \delta^{13}C_{phytoplankton} + [POC]_{terrestrial} \times \delta^{13}C_{terrestrial}\right) / [POC]_{sample}$$

$$[POC]_{phytoplankton} = [chl \ a]_{sample} \times (POC/chl \ a)_{mean}$$
 (eq. 2)

$$[POC]_{terrestrial} = [POC]_{sample} - [POC]_{phytoplankton}$$
 (eq. 3)

- where (POC/chl a)<sub>mean</sub> is the mean POC/chl a ratio of the samples used to determine
- phytoplankton signatures. Similar equations are used for the N/C ratio,  $\delta^{15}$ N and C/N ratio, but
- using PN instead of POC for  $\delta^{15}$ N and C/N ratio.
- Elemental and isotopic signatures of terrestrial POM can be estimated by directly measuring
- elemental and isotopic ratios in a sample like soil, rocks or vascular plants (e.g., Sarma et al.,
- 2014). However, this does not take into account the reworking of this material within the river
- system, which can affect these signatures (Hou et al., 2021). Thus, similarly to phytoplankton,
- elemental and isotopic signatures of terrestrial POM can be estimated using subsets of bulk
- data, following the approach of Savoye et al. (2012), Liénart et al. (2017) and Ferchiche et al.
- (2025, 2024). Labile terrestrial POM is usually characterised by high POC/chl a and C/N ratios
- and low POC/SPM ratios (Etcheber et al., 2007; Savoye et al., 2003 and references therein).
- However, during its decay in aquatic systems, terrestrial POM is colonised by bacteria (low
- C/N ratio), resulting in a consortium terrestrial POM + bacteria of lower C/N ratio than the
- original terrestrial POM (Etcheber et al., 2007; Savoye et al., 2012). Finally, one can
- discriminate two kinds of terrestrial POM: refractory terrestrial POM, characterised by high
- POC/chl a and C/N ratios and very low POC/SPM ratio, and quite labile terrestrial POM
- characterised by high POC/chl a ratio, intermediate C/N ratios and low POC/SPM ratio
- (Etcheber et al., 2007; Savoye et al., 2012). Thus, subsets of high POC/chl a ratio can be
- selected to determine the elemental and isotopic signatures of terrestrial POM. The C/N ratio
- can be used to discriminate labile from refractory terrestrial POM. When no samples exhibit a
- high POC/chl a ratio, samples exhibiting the highest (even if quite low) POC/chl a ratio can be
- used, but the data should first be corrected for the contribution of the phytoplankton POM using
- Equations 1-3.
- Elemental and isotopic ratios of riverine POM can exhibit a departure from a simple
- phytoplankton-terrestrial POM mixing. In the present study, this was the case in only two rivers.
- For the Têt River, the elemental and isotopic signature of anthropogenic POM was available in
- Higueras et al. (2014). It consisted of analyses of POM sampled in the wastewater treatment
- plant (WWTP) closest to the sampling site. For the Orb River, the signatures were estimated
- using the sample exhibiting the lowest  $\delta^{15}$ N, typical of anthropogenic POM (Ke et al., 2019).
- The estimation of POM-source signatures was performed independently for each river, except
- for some of the tributaries of the Arcachon Lagoon (rivers 11 to 15), where data sets were
- gathered, thanks to very similar characteristics (same δ13C of dissolved inorganic carbon;
- Polsenaere et al., 2013), to get a larger subset of data for estimating elemental and isotopic
- signatures more accurately. All criteria used for defining the above-described subsets are
- reported in Table 2.
- Table 2 Elemental and isotopic signatures of POM sources and criteria used to choose the data
- subset to determine them. When the signature did not vary over time, average  $\pm$  standard

deviation are reported. When the signature did vary over time, minimum and maximum values, standard deviations, as well as equations are reported. The types of mixing models performed for each river are also indicated (carbon mixing models were performed using  $\delta^{13}$ C and N/C ratio, or only  $\delta^{13}$ C; nitrogen mixing models were performed using  $\delta^{15}$ N and C/N ratio; mixed mixing models were performed using  $\delta^{13}$ C,  $\delta^{15}$ N and N/C ratio). POC% (or PN%) = Particulate Organic Carbon (or Particulate Nitrogen) to Suspended Particulate Matter ratio (%); C/N = POC/PN ratio (mol/mol); chla = chlorophyll a (µg/l); phaeo = phaeopigments (µg/l); conduc = conductivity (µS); temp = water temperature (°C); Q7 = mean of past seven days river flow; NO<sub>3</sub><sup>-</sup> = nitrate (mg(NO<sub>3</sub><sup>-</sup>)/l).

| -                                             |                                    | Source                              | discriminants                                                            |                      | Mo     | del performe | i     | L              | abile terre                                     | strial mat                                       | tter                                                | Refi                     | actory ter     | restrial n   | natter                                              |
|-----------------------------------------------|------------------------------------|-------------------------------------|--------------------------------------------------------------------------|----------------------|--------|--------------|-------|----------------|-------------------------------------------------|--------------------------------------------------|-----------------------------------------------------|--------------------------|----------------|--------------|-----------------------------------------------------|
| River                                         | Labile<br>terrestrial<br>matter    | Refractory<br>terrestrial<br>matter | Phytoplankton                                                            | WWTP's<br>POM        | Carbon | Nitrogen M   | lixed | $\delta^{13}C$ | $\delta^{15}N$                                  | C/N                                              | N/C                                                 | $\delta^{13} \mathrm{C}$ | $\delta^{15}N$ | C/N          | N/C                                                 |
| Seine                                         | C/N > 10                           |                                     | POC/chla < 200                                                           |                      | X      | X            |       | -28.5<br>± 0.3 | 6.6<br>± 0.9                                    | 10.6<br>± 0.3                                    | 0.093<br>± 0.002                                    |                          |                |              |                                                     |
| Orne                                          | C/N > 11                           |                                     | POC/chla < 500                                                           |                      | X      | X            |       | -28.4<br>± 0.3 | 5.8<br>± 1.0                                    | 12.4<br>± 0.4                                    | $0.082 \pm 0.003$                                   |                          |                |              |                                                     |
| Rance                                         | POC/chla ><br>200 and chla<br>< 10 |                                     | POC/chla < 150                                                           |                      | X      | X            |       | -26.8<br>± 0.2 | 6.1<br>± 0.7                                    | $\begin{array}{c} 8.8 \\ \pm \ 0.4 \end{array}$  | $\begin{array}{c} 0.113 \\ \pm 0.007 \end{array}$   |                          |                |              |                                                     |
| Elorn                                         | C/N > 12                           |                                     | POC/chla < 200                                                           |                      | X      | X            |       | -28.4<br>± 0.7 | 5.8<br>± 0.9                                    | 13.0<br>± 0.8                                    | $0.077 \pm 0.005$                                   |                          |                |              |                                                     |
| Aulne                                         | C/N > 11                           |                                     | POC/chla < 200<br>and C/N < 9                                            |                      | X      | X            |       | -28.9<br>± 0.8 | $\begin{array}{c} 5.8 \\ \pm \ 0.8 \end{array}$ | 12.1<br>± 1.1                                    | $\begin{array}{c} 0.08 \\ \pm 0.008 \end{array}$    |                          |                |              |                                                     |
| Loire                                         | POC/chla > 500                     |                                     | POC/chla < 200                                                           |                      | X      | x            |       | -28.1<br>± 0.1 | $5.9 \\ \pm 0.3$                                | $\begin{array}{c} 10.3 \\ \pm \ 0.2 \end{array}$ | $\begin{array}{c} 0.097 \\ \pm \ 0.002 \end{array}$ |                          |                |              |                                                     |
| Sèvre Niortaise                               | C/N > 14                           |                                     | POC/chla < 300                                                           |                      | X      |              |       | -28.0<br>± 0.4 |                                                 |                                                  | $0.057 \pm 0.040$                                   |                          |                |              |                                                     |
| Charente                                      | C/N > 12                           |                                     | POC/chla < 300                                                           |                      | X      | X            |       | -29.0<br>± 0.4 | 4.7<br>± 0.2                                    | 14.5<br>± 0.5                                    | 0.069<br>± 0.002                                    |                          |                |              |                                                     |
| Seudre                                        | POC/chla ><br>2000 and C/N<br>> 12 |                                     | POC/chla < 1000                                                          |                      | X      |              |       | -28.5<br>± 0.1 | - 0.2                                           | - 0.5                                            | - 0.002                                             |                          |                |              |                                                     |
| Porge                                         | C/N > 15                           |                                     | δ <sup>13</sup> C : POC/chla <<br>100 ; N/C : mean of<br>Cirès to Landes |                      | X      |              |       | -26.5<br>± 1.1 |                                                 |                                                  | $\begin{array}{c} 0.050 \\ \pm 0.007 \end{array}$   |                          |                |              |                                                     |
| Cirès / Renet /<br>Milieu / Lanton<br>/ Tagon | C/N > 15 and chla < 1              |                                     | POC/chla < 1000<br>and POC% > 10                                         |                      | X      |              |       | -28.5<br>± 0.5 |                                                 |                                                  | $\begin{array}{c} 0.053 \\ \pm \ 0.013 \end{array}$ |                          |                |              |                                                     |
| Leyre                                         | C/N > 15 and<br>chla < 1           |                                     | POC/Chla < 1000. $\delta^{13}$ C < 28.59 and POC% > 10                   |                      | X      |              |       | -28.3<br>± 0.5 |                                                 |                                                  | $\begin{array}{c} 0.06 \\ \pm 0.005 \end{array}$    |                          |                |              |                                                     |
| Landes                                        | C/N > 12                           |                                     | POC/Chla < 600. $\delta^{13}$ C < -29.1                                  |                      | X      |              |       | -29.1<br>± 0.4 |                                                 |                                                  | $0.075 \pm 0.002$                                   |                          |                |              |                                                     |
| Adour                                         | POC/chla > 3000                    |                                     | POC/chla < 200                                                           |                      | X      |              |       | -26.0<br>± 0.9 |                                                 |                                                  | 0.099<br>± 0.008                                    |                          |                |              |                                                     |
| Têt                                           | C/N > 11.5                         | POC% < 4.25                         | PN% > 2. $\delta^{13}$ C < 26 and $\delta$ 15N > 5                       | Measured             |        |              | X     | -26.0<br>± 0.2 | 3.7<br>± 0.6                                    | 12.2<br>± 0.5                                    | $\begin{array}{c} 0.082 \\ \pm 0.002 \end{array}$   | -26.0<br>± 0.6           | 6.7<br>± 1.4   | 5.8<br>± 1.4 | $\begin{array}{c} 0.180 \\ \pm \ 0.045 \end{array}$ |
| Aude                                          | C/N > 12                           | Q7 > 70                             | PN% > 1 or 2 and<br>C/N < 6                                              |                      |        |              | X     | -28.1<br>± 0.6 | 6.3<br>± 0.1                                    | 15.3<br>± 1.6                                    | 0.066<br>± 0.007                                    | -28.0<br>± 0.7           | 4.7<br>± 0.4   | 7.3<br>± 1.0 | 0.139<br>± 0.018                                    |
| Orb                                           | C/N > 10                           |                                     | $PN\% > 2. \delta^{15}N > 4.06$                                          | Lower $\delta^{15}N$ |        |              | X     | -27.1<br>± 0.4 | 3.7<br>± 0.4                                    | 10.5<br>± 0.3                                    | $0.095 \pm 0.350$                                   |                          |                |              |                                                     |
| Hérault                                       | C/N > 12                           | Q > 45                              | PN% > 2                                                                  |                      |        |              | X     | -27.7<br>± 0.2 | 6.1<br>± 0.7                                    | 13.7<br>± 1.2                                    | 0.073<br>± 0.007                                    | -27.8<br>± 0.4           | 4.7<br>± 0.6   | 8.2<br>± 1.5 | 0.124<br>± 0.019                                    |
| Rhône                                         | C/N > 12                           | POC% < 1.25                         | $C/N < 6.68$ and $\delta^{15}N > 3.92$                                   |                      |        |              | X     | -26.4<br>± 1.3 | 5.2<br>± 1.0                                    | 17.0<br>± 3.2                                    | $0.061 \pm 0.012$                                   | -25.9<br>± 0.4           | 3.1<br>± 0.8   | 8.8<br>± 3.1 | 0.119<br>± 0.032                                    |

#### Table 2 (continued)

| -                                             |                        |                                                                                               | Pł                                              | ytoplankton                                                     |                                                 |                                                     |                                      |                | WWTI           | 's POM                                          |                                                   |
|-----------------------------------------------|------------------------|-----------------------------------------------------------------------------------------------|-------------------------------------------------|-----------------------------------------------------------------|-------------------------------------------------|-----------------------------------------------------|--------------------------------------|----------------|----------------|-------------------------------------------------|---------------------------------------------------|
| River                                         | $\delta^{13}$ C        | + equations                                                                                   | $\delta^{15}N$                                  | + equations                                                     | C/N                                             | N/C +                                               | equations                            | $\delta^{13}C$ | $\delta^{15}N$ | C/N                                             | N/C                                               |
| Seine                                         | -32.8<br>± 1.1         |                                                                                               | 8.4<br>± 1.7                                    |                                                                 | 7.4<br>± 0.7                                    | 0.136<br>± 0.012                                    |                                      |                |                |                                                 |                                                   |
| Orne                                          | -31.4<br>± 0.8         |                                                                                               | $\begin{array}{c} 4.3 \\ \pm \ 0.8 \end{array}$ |                                                                 | 6.6<br>± 1.3                                    | $0.141 \pm 0.010$                                   |                                      |                |                |                                                 |                                                   |
| Rance                                         | [-31.4;-25;6]<br>± 1.7 | 5.7x10 <sup>-4</sup> ×[chla+phaeo] <sup>2</sup> -<br>0.04×[chla+phaeo]-30.6                   | [4.7;11.4]<br>± 0.7                             | -0.28×[NO <sub>3</sub> <sup>-</sup> ]+12.7                      | $\begin{array}{c} 6.2 \\ \pm \ 0.4 \end{array}$ | $\begin{array}{c} 0.161 \\ \pm 0.010 \end{array}$   |                                      |                |                |                                                 |                                                   |
| Elorn                                         | -27.4<br>± 0.3         |                                                                                               | 6.9<br>± 0.5                                    |                                                                 | $10.0 \\ \pm 0.9$                               | $\begin{array}{c} 0.101 \\ \pm 0.007 \end{array}$   |                                      |                |                |                                                 |                                                   |
| Aulne                                         | -28.1<br>± 0.2         |                                                                                               | $\begin{array}{c} 8.6 \\ \pm \ 0.2 \end{array}$ |                                                                 | $\begin{array}{c} 8.2 \\ \pm \ 0.2 \end{array}$ | $0.122 \pm 0.003$                                   |                                      |                |                |                                                 |                                                   |
| Loire                                         | [-30.6;-25.0]<br>± 0.9 | 5x10 <sup>-4</sup> ×[chla+phaeo] <sup>2</sup> -<br>0.02[chla+phaeo]-<br>0.39[chla/phaeo]-27.9 | [3.0;10.4]<br>± 1.2                             | 4.2x10 <sup>-4</sup> [chla] <sup>2</sup> -<br>0.08[chla]+8.2    | $\begin{array}{c} 7.2 \\ \pm \ 0.6 \end{array}$ | $\begin{array}{c} 0.140 \\ \pm 0.011 \end{array}$   |                                      |                |                |                                                 |                                                   |
| Sèvre Niortaise                               | [-35.7;-29.2]<br>± 1.0 | -258×exp([chla+phaeo]²/<br>16055)-0.15×[temp]+229                                             |                                                 |                                                                 |                                                 | $[0.106; 0.145] \\ \pm 0.006$                       | 2.9x10 <sup>-3</sup> ×[chla+phaeo] + |                |                |                                                 |                                                   |
| Charente                                      | -30.8<br>± 0.03        |                                                                                               | 7.5<br>± 1.6                                    |                                                                 | $6.6 \\ \pm 0.3$                                | $0.152 \pm 0.006$                                   |                                      |                |                |                                                 |                                                   |
| Seudre                                        | -33.3<br>± 0.1         |                                                                                               |                                                 |                                                                 |                                                 |                                                     |                                      |                |                |                                                 |                                                   |
| Porge                                         | -33.6<br>± 0.4         |                                                                                               |                                                 |                                                                 |                                                 | $\begin{array}{c} 0.128 \\ \pm 0.008 \end{array}$   |                                      |                |                |                                                 |                                                   |
| Cirès / Renet /<br>Milieu /<br>Lanton / Tagon | -34.9<br>± 0.4         |                                                                                               |                                                 |                                                                 |                                                 | $\begin{array}{c} 0.133 \\ \pm \ 0.006 \end{array}$ |                                      |                |                |                                                 |                                                   |
| Leyre                                         | -30.1<br>± 0.3         |                                                                                               |                                                 |                                                                 |                                                 | $\begin{array}{c} 0.140 \\ \pm \ 0.016 \end{array}$ |                                      |                |                |                                                 |                                                   |
| Landes                                        | -29.9<br>± 0.3         |                                                                                               |                                                 |                                                                 |                                                 | $0.112 \pm 0.010$                                   |                                      |                |                |                                                 |                                                   |
| Adour                                         | -28.2<br>± 0.6         |                                                                                               |                                                 |                                                                 |                                                 | $0.111 \pm 0.010$                                   |                                      |                |                |                                                 |                                                   |
| Têt                                           | [-29.7;-27.8]<br>± 0.6 | -5.2×10 <sup>-3</sup> [temp] <sup>2</sup><br>+0.08×[temp]-27.5                                | [5.3;13.3]<br>± 1.8                             | 5.53×[temp]-5.5                                                 | $\begin{array}{c} 5.6 \\ \pm \ 0.7 \end{array}$ | $0.181 \pm 0.021$                                   |                                      | -26.3<br>± 0.1 | -0.7<br>± 0.1  | $\begin{array}{c} 6.3 \\ \pm \ 0.3 \end{array}$ | $\begin{array}{c} 0.160 \\ \pm 0.017 \end{array}$ |
| Aude                                          | [-32.6;-27.8]<br>± 0.6 | -0.21×[temp]-26.5                                                                             | [5.2;10.6]<br>± 1.6                             | $-1.13 \times \delta^{13}$ C-26.2                               | $5.0 \\ \pm 0.8$                                | $0.205 \pm 0.033$                                   |                                      |                |                |                                                 |                                                   |
| Orb                                           | [-30.7;-23.4]<br>± 0.6 | -0.19×[temp]-26.0                                                                             | [4.9;8.4]<br>$\pm 0.6$                          | 8.44-(3.63×(conduc-<br>505))/(conduc-111)                       | $\begin{array}{c} 4.8 \\ \pm 0.9 \end{array}$   | $0.213 \pm 0.039$                                   |                                      | -27.1<br>± 0.4 | 1.9<br>± 1.9   | $\begin{array}{c} 3.7 \\ \pm 3.7 \end{array}$   | $\begin{array}{c} 0.270 \\ \pm 0.270 \end{array}$ |
| Hérault                                       | [-31.5;-27.5]<br>± 1.0 | -0.19×[temp]-26.0                                                                             | [6.3;10;9]<br>± 1.3                             | 3.6x10 <sup>-2</sup> ×[temp] <sup>2</sup> -<br>1.15×[temp]+14.6 | 5.0<br>± 0.7                                    | $0.203 \pm 0.031$                                   |                                      |                |                |                                                 |                                                   |
| Rhône                                         | -27.8<br>± 1.2         |                                                                                               | 5.6<br>± 0.8                                    |                                                                 | $5.5 \\ \pm 0.8$                                | $0.180 \pm 0.030$                                   |                                      |                |                |                                                 |                                                   |

### 2.4. Quantification of POM composition

POM composition was quantified using a Bayesian mixing model ('simmr' R package version 0.4.5, Govan and Parnell, 2023), which solves the equations system based on bulk and source POM elemental and isotopic signatures. Mixing models were computed for each sampling date of each river (Tab. 1), using carbon ( $\delta^{13}$ C and N/C ratio, Eq. 4, 7, 8), nitrogen ( $\delta^{15}$ N and C/N ratio, Eq. 5, 6, 8), and/or a combination of three ( $\delta^{13}$ C,  $\delta^{15}$ N and N/C ratio, Eq. 4, 5, 7, 8) tracers. From the three mixing models performed for each sampling date and river (carbon, nitrogen or mixed), one model was selected as the best estimation of bulk POM data. It should be noted that N/C and C/N ratios give information on the mixing of C and N, respectively (Perdue and Koprivnjak, 2007). We used at least the same number of equations as unknowns (sources) to avoid running underdetermined models that result in large uncertainty in model outputs(Phillips et al., 2014). Equations of the models were:

$$281 \qquad \delta^{13}C_{\text{mixture}} = x_1 \ \delta^{13}C_{\text{source 1}} + x_2 \ \delta^{13}C_{\text{source 2}} + x_3 \ \delta^{13}C_{\text{source 3}} + x_4 \ \delta^{13}C_{\text{source 4}} \tag{Eq. 4}$$

$$\delta^{15}N_{mixture} = x_1 \ \delta^{15}N_{source \ 1} + x_2 \ \delta^{15}N_{source \ 2} + x_3 \ \delta^{15}N_{source \ 3} + x_4 \ \delta^{15}N_{source \ 4} \tag{Eq. 5}$$

$$C/N_{\text{mixture}} = x_1 C/N_{\text{source } 1} + x_2 C/N_{\text{source } 2} + x_3 C/N_{\text{source } 3} + x_4 C/N_{\text{source } 4}$$
 (Eq. 6)

$$N/C_{\text{mixture}} = x_1 N/C_{\text{source } 1} + x_2 N/C_{\text{source } 2} + x_3 N/C_{\text{source } 3} + x_4 N/C_{\text{source } 4}$$
(Eq. 7)

$$x_1 + x_2 + x_3 + x_4 = 1$$
 (Eq. 8)

As there was no *a priori* knowledge of sources contributions to the POM mixture, the models were set with an uninformative prior (1, 1, 1, 1) following a Dirichlet distribution (all sources have an equal probability to contribute to the mix; Phillips et al., 2014). Model runs were set following the recommendations of Phillips et al. (2014). Models outputs were evaluated with Gelman-Rubin diagnostic (verification of chain convergence) and predictive distributions to ensure the good fit of the models to the observed data. Models outputs are given as medians.

Absolute uncertainties for the models varied from 1 to 18 % (range of average for each river)

with an overall average of 8 % (all models).

#### 2.5. Forcings at local and multi-system scales

Environmental forcings driving POM composition were determined using redundancy analysis (RDA; 'dudi.pca' and 'pcaiv' functions; R package {ade4} version 1.7-19). RDA summarises multiple linear regressions between the response variable (POM composition: mixing model outputs) and a set of explanatory variables (environmental forcings) to assess causality links (Legendre et al., 2011). RDAs were performed at single-river and multi-river scales. Regarding the multi-river scale, the annual mean POM composition of each river was used to determine the drivers of spatial (i.e., between-rivers) variations of POM composition.

The proxies of the environmental forcings were chosen to directly or indirectly reflect the forcings that affect the processes occurring in the river and the adjacent ecosystems and influencing POM source inputs. To homogenise the data sets for running the single-river RDAs, the same combination of twelve parameters (see Table A2) was used for each river. They are linked to primary production (chlorophyll *a*, phaeopigments, temperature, pH, ammonium, nitrate, phosphate, irradiance), upstream and lateral, natural and/or anthropogenic inputs (river flow, rain, SPM, ammonium, nitrate, phosphate), and resuspension (SPM, zonal and meridional wind energy). For the multi-river RDA, environmental proxies were selected to reflect processes occurring at large spatial scales and in the river basin. Forty parameters (See Fig. A6) were used. They are linked to water quality (conductivity, nitrates), climate setting (river flow, latitude, longitude, air temperature, precipitation, zonal, meridional and total wind energy), geomorphology (river length and flow, basin surface area and slope, Strahler number), land use coverage (agricultural areas, artificial surfaces, forest and seminatural areas, wetlands and water bodies), catchment soil properties (organic carbon content, net erosion, granulometry, useful

water reserves<sup>1</sup>), soil type (acidic, brown, organic and hydromorphic soil), bedrock type (alluvial/unconsolidated deposits, calcareous and marl rocks, clayey materials, detrital formations, sandy materials, loess, crystalline and metamorphic rocks, volcanic rocks and other/organic materials), and urban pressure (WWTP capacities, WWTP capacities to river flow ratio). From this initial list of proxies, some were removed to limit the auto-correlation (use of the Variance Inflation Factor, Borcard et al., 2011) and to improve the adjusted R<sup>2</sup> of each RDA analysis (Tab. A2 and Fig. A6).

#### 2.6. Typology of river dynamics

Rivers were classified based on POM composition and their temporal dynamics by performing a regionalisation analysis as in Liénart et al. (2018) (Fig. A1). This method, based on multivariate cluster analysis (Souissi et al., 2000), allows to consider the temporal (seasonal) variations specific to each river in addition to the spatial (between-rivers) component. The regionalisation analysis was based on POM composition data (i.e., proportions of sources) computed for each river and each month. When the sampling was fortnightly, averages were performed to get one value per month. When more than one year was sampled, a standard year was chosen. Nevertheless, to check if the choice of one year over the other ones would modify the typology, another regionalisation was performed using all available years for all rivers. This regionalisation also allowed to check if a river can shift from one type to another type depending on year and associated environmental conditions (inter-annual variability of type belonging). Also, in order to check if the over-representation of the small rivers and streams fuelling the Arcachon Bay would bias the typology, a third regionalisation was performed, reducing the numbers of these rivers from 8 to 3 (and especially from 6 to 1 regarding rivers of Type I). The results (Fig. 5, Fig. A7) are very similar, indicating the robustness of the method.

A contingency matrix (rivers, sources, months) was created from monthly values of source contributions (i.e., mixing model outputs). For each month, a dendrogram was performed, and ten cut-off levels were considered. Then, for each cut-off level, similarities between stations were identified within the twelve-monthly dendrograms. Ultimately, global similarities between rivers were computed using a fuzzy cluster that returns probabilities of membership of each river to each cluster type. The best number of river types, i.e., river dynamics typology, was determined considering the best Dunn coefficient (Dunn, 1974) and Silhouette score (Rousseeuw, 1987).

#### 3. Results

Hereafter, four rivers (Rance, Charente, Milieu and Hérault Rivers) were selected and considered as representative of each type of studied river (see section 3.4). Thus, most of the

<sup>&</sup>lt;sup>1</sup> Useful water reserve corresponds to the water retained by the soil and that is useful for plants. It is calculated from soil thickness and granulometry (Le Bas, 2025)

results are illustrated using these four rivers. Graphs of all the other rivers are reported in the supplementary material.

#### 3.1. Contrasting seasonalities in river characteristics

351

352

353 As stated in section 2.1, the 23 studied rivers encompassed large gradients of environmental 354 characteristics, as illustrated by the lowest and highest annual means of river flow (0.3 and 1572 355 m<sup>3</sup>/s; Lanton and Rhône Rivers), water temperature (12.3 to 17.1 °C; Cirès and Têt Rivers), SPM (2.7 and 40.9 mg/l; Cirès and Rhône River), POC (0.3 and 5.1 mg/l; Hérault and Loire 356 357 Rivers) and chlorophyll a (0.4 to 57.1 µg/l; Cirès and Rance Rivers) concentrations as well as POC/chl a (199 and 6444 g/g; Loire and Leyre Rivers) and C/N (5.9 and 20.3 mol/mol; Têt and 358 Lanton Rivers) ratios; this was less contrasting among rivers for  $\delta^{13}$ C (-30.2 and -26.2 %; Sèvre 359 and Têt Rivers) and especially  $\delta^{15}N$  (4.0 and 8.0 ‰; Leyre and Rance Rivers) (Fig. 2, A2). 360 As generally observed in rivers from mid-latitude, the studied rivers exhibited clear seasonal 361 362 patterns in water temperature with lower and higher values in winter and summer, respectively. 363 However, such clear seasonal patterns were not always recorded for all the parameters, as there 364 were contrasting patterns of seasonal variability among rivers. Indeed, the seasonal variability 365 of river flow was quite smooth (e.g., the Rance and Charente Rivers) with a higher flow in winter/spring and lower flow in summer/fall for some rivers, whereas it was highly pulsed for 366 367 some others with constant low levels marked by short and strong floods (e.g., 53m<sup>3</sup>/s in mean 368 but 1169m<sup>3</sup>/s in flood time for the Hérault River) (Fig. 2). Overall, one can distinguish rivers 369 that are characterized by high concentrations of chlorophyll a and clear seasonal patterns of 370 most parameters (e.g., 53 µg/l of chlorophyll a in mean ranging from 3 to 135 µg/l in the Rance 371 River) from rivers characterized by low concentrations of chlorophyll a, high POC/chl a and 372 low seasonal variability for most of the parameters (e.g., 1.1 µg/l of chlorophyll a in mean ranging from 0.7 to 1.7 µg/l in the Milieu River) and from rivers that are characterized by high 373 374 seasonal variability of most parameters but without a clear seasonal pattern (e.g., Hérault 375 River). Other rivers exhibited intermediate behaviour (e.g., Charente River) (Fig. 2, A2). 376 Usually, Rance-like rivers exhibited high concentrations of chlorophyll a in spring/summer 377 associated with POC/chl a ratio lower than 200 g/g, C/N ratio lower than 8 mol/mol and low δ<sup>13</sup>C (down to -31 ‰ or -33 ‰; e.g., Seine River, Fig. A2). In contrast, Milieu-like rivers 378 379 exhibited high POC/chl a > 700 g/g) and C/N ratio (> 15 mol/mol) and quite constant  $\delta^{13}$ C 380  $(\sim -29 - -28 \%)$  all year round (e.g., Cirès and Renet Rivers). These rivers are tributaries of the 381 Arcachon Lagoon. Hérault-like rivers flowing into the Mediterranean Sea exhibited highly and suddenly variable C/N ratios (4 – 17 mol/mol),  $\delta^{13}$ C (~-33 – -26 ‰) and  $\delta^{15}$ N (~2 – 12 ‰) (e.g., 382 Aude and Orb Rivers; Fig. A2). 383

- <u>Figure 2</u> Temporal variations of matter characteristics for representative rivers along the studied
- periods for  $\delta^{13}$ C (left axis; black dotted line) and  $\delta^{15}$ N (right axis; blue line) (first column); C/N
- (left axis; black dotted line) and POC/chl a (right axis; blue line) ratios (second column);
- SPM(left axis; black dotted line), POC (right axis; blue line) and chl a (right axis; blue dotted
- line) concentrations (third column) and river flow (left axis; black dotted line) and temperature
- (right axis; blue line) (fourth column).

#### 3.2. Elemental and isotopic signatures of POM sources

- Elemental and isotopic signatures of phytoplankton were estimated for each of the twenty-three
- rivers (Tab. 2, Fig. 3 and A3). Most of them (all of them for the C/N ratio) were found to be
- constant over time. Their annual mean values varied between -34.9 % (some tributaries of the
- Arcachon Lagoon) and -27.4 % (Elorn River) for  $\delta^{13}$ C, between 4.3 % (Elorn River) and 8.6
- 396 % (Aulne River) for  $\delta^{15}$ N and between 4.8 mol/mol (Orb River) and 10.0 mol/mol (Elorn River)
- for the C/N ratio. Some of them varied over time along with pigment concentration and ratio or
- with temperature for  $\delta^{13}$ C, and with pigment concentration (chlorophyll a and/or
- phaeopigments), nitrate concentration, temperature,  $\delta^{13}$ C or conductivity for  $\delta^{15}$ N (Tab. 2). The
- range of temporal variability was usually 4-6 % for  $\delta^{13}$ C and  $\delta^{15}$ N. Overall, phytoplankton
- signatures are comprised between -35.6 and -23.8 % for the  $\delta^{13}C$  and between 3.0 and 13.2 %
- for the  $\delta^{15}$ N.

- All other signatures were found to be constant over time (Tab. 2 and A2, Fig. 3 and A3) but
- may differ between rivers. Signatures mean annual values of labile terrestrial POM were
- comprised between -29.1 and -26.0 % for the  $\delta^{13}$ C, between 3.7 and 6.6 % for the  $\delta^{15}$ N and
- between 8.8 and 17.0 mol/mol for the C/N ratio. Signatures mean annual values of refractory
- terrestrial POM were comprised between -28.0 and -25.9 % for the  $\delta^{13}$ C, between 3.1 and 6.7
- 408 % for the  $\delta^{15}$ N and between 5.8 and 8.8 mol/mol for the C/N ratio. Signatures mean annual
- values of sewage POM were -27.1 and -26.3 % for  $\delta^{13}$ C, 1.9 and -0.7 % for  $\delta^{15}$ N and 3.7 and
- 6.3 mol/mol for C/N ratio for Orb and Têt Rivers, respectively.

Figure 3  $\delta^{13}$ C,  $\delta^{15}$ N, N/C or C/N values of bulk POM (black crosses) and sources. The latter are presented as closed circles (average) and bars (standard deviation) when the signatures were constant over time and by colored area when at least one of the proxies was variable over time (see Table 2). This colored area corresponds to the dispersion of the values, including their uncertainties.

#### 3.3. Dynamics of particulate organic matter composition

Particulate organic matter composition resulting from mixing models outputs is presented 419 hereafter, for each river, as the relative contribution of each source to the POM pool (Fig. 4). 420 Among rivers whose POM is composed of only two sources (terrestrial POM and 421 phytoplankton), one can distinguish rivers with terrestrial-dominated POM (e.g., Milieu River: 422 terrestrial POM accounted for  $94 \pm 3$  % of the mixture) to rivers of intermediate POM 423 composition (e.g., Charente and Rance Rivers where phytoplankton accounted for  $34 \pm 10$  % 424 and  $62 \pm 10$  % of the mixture, respectively). All these rivers flow into the English Channel and 425 the Atlantic Ocean. The rivers whose POM is composed of three or four sources flow into the 426 Mediterranean Sea. In these rivers, terrestrial POM is present as refractory and labile materials. 427 The contribution of labile terrestrial POM ranged between  $16 \pm 15$  % (Têt River) and  $46 \pm 21$ 428 % (Orb River), and of refractory terrestrial POM between  $21 \pm 9$  % (Rhône River) and  $39 \pm 15$ 429 % (Aude River). The contribution of phytoplankton ranged between  $34 \pm 15$  % (Aude River) 430 and  $51 \pm 30$  % (Hérault River) for the Mediterranean rivers. The fourth source of POM was the 431 WWTP's POM. It was identified as a source in the Orb and Têt Rivers and accounted for  $15 \pm$ 432 6 % and  $10 \pm 7$  % in these two rivers, respectively. Regarding temporal variations of POM 433 composition, some rivers exhibited clear seasonal patterns, whereas others revealed a 434 homogeneous composition over the annual cycle (Fig. 4). The rivers where POM was highly 435 dominated by terrestrial POM (Seudre, Cirès, Renet, Lanton, Milieu, Tagon, Leyre Rivers) 436 showed almost no seasonal variability. In contrast, some rivers like the Rance, the Elorn or the 437 Aulne River showed a clear seasonal pattern with the dominance of terrestrial material in winter 438 and phytoplankton in summer. At last, other rivers exhibited less clear (e.g., Landes, Porge, 439 Charente Rivers) or even no clear seasonal pattern but a quite stochastic variability over the 440 annual cycle (e.g., Sèvre, Adour, Aude, Orb).

It should be noted that the above is valid for carbon and mixed as well as nitrogen models (cf.

Tab. 2; Fig. 4 and A4).

<u>Figure 4</u> Temporal dynamic (rectangle graphs) and (inter-)annual mean (pie charts) of POC source proportions. Sources are phytoplankton (green), labile terrestrial material (brown), refractory terrestrial material (yellow) and anthropogenic POM (orange).

#### 3.4. Typology of river dynamics

452453

458459

467468

Four types of river dynamics were determined by the regionalisation analysis based on river POM compositions and their temporal dynamics (Fig. 5). The seven rivers (Renet, Cirès, Lanton, Milieu, Seudre, Tagon and Leyre River), mainly belonging to Type I, were characterised by terrestrial-dominated POM and no/low seasonality. Six of them are small streams/rivers flowing to the Arcachon Lagoon. The five rivers (Aude, Hérault, Têt, Rhône and Orb River), mainly belonging to Type II, were characterised by the co-occurrence of labile and refractory terrestrial POM and large temporal variability, but, except for the Hérault River, without a clear seasonal pattern. They all flow to the Mediterranean Sea. The five rivers (Porge, Adour, Charente, Orne and Landes River), mainly belonging to Type III, were composed of phytoplankton and terrestrial POM, and exhibited moderate seasonality. Type III is clearly an intermediary between Type I and Type IV. These five rivers flow to the Atlantic Ocean or the English Channel. Among the seven rivers flowing to the Arcachon Lagoon, the two that mainly belong to Type III are man-managed streams and flow through lakes, contrary to the six other ones, which mainly belong to Type I and are natural streams that do not flow through lakes. Finally, the six rivers (Rance, Elorn, Aulne, Loire, Seine and Sèvre River) mainly belonging to Type IV were composed of phytoplankton and terrestrial POM, and exhibited high seasonality. These six rivers flow to the Atlantic Ocean or the English Channel. It should be noted that the regionalisations performed using all sampled years for all rivers (Fig. A7) resulted in the same typology and in the same type for each river, whatever the sampling year. The only exception is the Leyre River, which switched from Type III in 2008 to Type I in 2009.

Figure 5 Typology of river dynamics following a hierarchical cluster analysis on POM source proportions. The percentages of membership for each type attributed to each river are shown.

#### 3.5. Environmental forcings driving POM composition

One redundancy analysis was performed for each river to relate environmental parameters, considered as proxies of drivers, to the POM composition, i.e., to assess the drivers of the temporal variability of POM composition for each river (Fig. 6 and A5). It should be kept in mind that the POC or PN concentration of each source was used for these analyses and not the relative proportion of the sources. In type-I rivers, i.e., rivers characterised by terrestrialdominated POM and no/low seasonality, terrestrial POM is usually linked to river flow and/or SPM concentration (e.g., Milieu River on Fig. 6, Leyre and Tagon Rivers in Fig. A5). However, this feature is not always clear since the POM of these rivers is always dominated by terrestrial material, regardless of the environmental conditions. In type-II rivers, i.e., rivers characterised by the co-occurrence of labile and refractory terrestrial POM and large temporal variability, phytoplankton POM is usually positively linked to temperature and negatively linked to river flow, whereas labile and refractory terrestrial POM are both positively linked to SPM and/or river flow. Precisely, labile terrestrial POM is usually better linked to river flow and refractory terrestrial POM to SPM (e.g., Hérault River in Fig. 6 and Rhône River in Fig. A5). In the Têt River, anthropogenic POM was linked to nitrate concentration (Fig. A5). In rivers characterised by phytoplankton and terrestrial-POM composition with moderate (Type III) or high (Type IV) seasonality, terrestrial POM was almost always positively linked to river flow and/or SPM concentration, while phytoplankton was usually linked with chlorophyll a concentration (e.g., Charente and Rance Rivers on Fig. 6, Landes and Seine Rivers on Fig. A5).

Figure 6 Redundancy analyses (correlation circles) of rivers standing for each type of river. Black arrows represent explained variables (concentration of POC sources) and red arrows represent explaining variables (environmental variables). River types are recalled (Roman numerals). LT<sub>POC</sub> = Labile terrestrial POC; RT<sub>POC</sub> = Refractory terrestrial POC;  $\varphi_{POC}$  = Phytoplankton POC; Chl a = chlorophyll a; Phaeo. = phaeopigments; M. wind = meridional wind; Z. wind = zonal wind; R. flow = river flow; Temp. = temperature; Irrad. = Irradiance; NH<sub>4</sub><sup>+</sup> = ammonium; NO<sub>3</sub><sup>-</sup> = nitrate; PO<sub>4</sub><sup>3-</sup> = phosphates; Adj. R<sup>2</sup> = adjusted R<sup>2</sup>.

At last, another RDA was performed, gathering all rivers to relate environmental parameters to the mean annual POM composition at the multi-river scale (Fig. 7). As anthropogenic POM was only detected in two rivers (Orb, Têt), it was not included in the multi-river analysis to avoid analysis bias. At this scale, phytoplankton is strongly positively correlated to agricultural surfaces and conductivity, labile terrestrial material to soil organic carbon content and acidic soil coverage, and refractory terrestrial material to catchment slope; refractory terrestrial material is also negatively correlated to soil useful water reserves(all correlations are

significant; Fig. A6). Note that the phytoplankton and labile terrestrial matter, as well as their related environmental variables, are negatively correlated.

<u>Figure 7</u> Multi-river redundancy analysis. Black arrows represent explained variables (relative proportions), red arrows represent explaining variables (environmental variables), and numbers are river identifiers (cf. Fig.1). R. flow = river flow; %OC soil = percentages of organic carbon in soil;  $NO_3^-$  = nitrates concentration; Useful reserve = Useful water reserve in soil; Conduc. = conductivity; %Agri. Surf. = Proportion of agricultural surface; %Acid. soil = Proportion of acidic soil coverage; Slope = Catchment slope; Adj.  $R^2$  = adjusted  $R^2$ .

#### 4. Discussion

#### 4.1. Bulk POM and source signatures in temperate rivers

Over the 23 studied rivers,  $\delta^{13}$ C,  $\delta^{15}$ N, and C/N ratios of bulk POM ranged between -35.2 and -24.5 ‰, -0.3 and 12.6 ‰, and 3 and 23.4 mol/mol, respectively. This corresponds to usual values recorded for riverine POM over temperate systems, except for the lowest C/N ratios (Ferchiche et al., 2024; Kendall et al., 2001; Ogrinc et al., 2008).

In the present study, isotopic and elemental signatures of terrestrial POM and phytoplankton were determined from subsets of the bulk data sets following the approaches of Savoye et al. (2012), Liénart et al. (2017) and Ferchiche et al. (2025, 2024). It has the double advantage of 1) taking into account the reworking of terrestrial POM within the river and thus discriminating labile from refractory terrestrial POM, and 2) taking into account the variability of phytoplankton signature over time, due to differences in growth conditions (see below). Labile terrestrial POM mainly appears during high river flow (Fig. 6 and A5; Savoye et al., 2012) and is usually composed of riparian litter (e.g., Veyssy et al., 1998). In the studied rivers,  $\delta^{13}$ C,  $\delta^{15}$ N and C/N ratio of labile terrestrial POM ranged between -28.9 ± 0.8 ‰ and -26 ± 0.9 ‰, 3.7 ± 0.6 ‰ and 6.6 ± 0.9 ‰, and 8.8 ± 0.4 and 17 ± 3.2 mol/mol, respectively. These values are very similar to values found in other temperate systems like the Gironde Estuary ( $\delta^{13}$ C = -28.7 ± 0.9

%; Savoye et al., 2012), the Sava River ( $\delta^{13}C = -28 \pm 5$  %;  $\delta^{15}N = 5 \pm 2$  %;  $C/N = 33 \pm 15$ 537 mol/mol. Ogrinc et al., 2008) or Taiwanese rivers ( $\delta^{13}C = -26.6 \pm 1.8$  %; C/N = 31.1 ± 23.4 538 mol/mol<sub>2</sub> Hilton et al., 2010) and very similar to direct measurement of C3 plants ( $\delta^{13}$ C = -28.1 539  $\pm 2.5$  %; O'Leary, 1981 and references therein;  $\delta^{13}C = -28 \pm 1.3$  %;  $\delta^{15}N = 0.8 \pm 2.9$  %; C/N 540 = 39.6  $\pm$  25.7 mol/mol; Dubois et al., 2012;  $\delta^{13}$ C = -27.9  $\pm$  0.1 %; Fernandez et al., 2003). 541 542 Refractory terrestrial POM is terrestrial POM that has undergone large reworking within river water, river sediment or even the estuarine maximum turbidity zone (e.g., Etcheber et al., 2007; 543 Veyssy et al., 1998). In the studied rivers where it was found,  $\delta^{13}$ C,  $\delta^{15}$ N and C/N ratios of 544 refractory terrestrial POM ranged between  $-28 \pm 0.7$  % and  $-25.9 \pm 0.4$  %,  $3.2 \pm 0.8$  % and 6.7545 546  $\pm$  1.4 ‰, and 5.8  $\pm$  1.4 and 8.8  $\pm$  3.1 mol/mol, respectively. These values are very similar to the large gradient of refractory POM origins found in other temperate systems like the Gironde 547 Estuary (France) (resuspended sediment,  $\delta^{13}C = -25.2 \pm 0.3$  %;  $\delta^{15}N = 5.5 \pm 0.4$  %; C/N = 8.5548  $\pm$  0.8 mol/mol; Savoye et al., 2012), Taiwanese rivers (petrogenic POM,  $\delta^{13}$ C = -23.6  $\pm$  1.1 ‰; 549  $C/N = 6.5 \pm 1.6$  mol/mol; Hilton et al., 2010) and in the Pearl River (China) (soil,  $\delta^{13}C$ : between 550  $-28.3 \pm 0.8$  % and  $-21.7 \pm 0.7$  %; C/N: between  $8.9 \pm 1.1$  and  $17.9 \pm 3.6$  mol/mol; Yu et al., 551 552 2010). Isotopic signatures of phytoplankton vary depending on biogeochemical conditions and 553 554 processes like nutrient availability and utilisation, growth rate and limitation (e.g., Fry, 1996; 555 Liénart et al., 2017; Lowe et al., 2014; Miller et al., 2013; Savoye et al., 2003; Sigman et al., 556 2009; Yan et al., 2022) and can be estimated using measured environmental parameters 557 (Ferchiche et al., 2024, 2025; Liénart et al., 2017; Savoye et al., 2012). For the seven rivers where phytoplankton isotopic signatures were found variable over time, phytoplankton  $\delta^{13}$ C or 558 559  $\delta^{15}$ N were correlated to: concentrations and ratio of chlorophyll a and phaeopigments, water temperature, nitrate concentration and/or conductivity (Tab. 2). Chlorophyll a and 560 phaeopigments concentrations are direct proxies of phytoplankton fresh and degraded 561 562 biomasses and are related to phytoplankton growth and decay, two processes that increase phytoplankton  $\delta^{13}$ C (Golubkov et al., 2020; Michener and Kaufman, 2007 and references 563 therein). Similar processes may explain phytoplankton- $\delta^{15}$ N increase with chlorophyll a 564 increase. An increase in water temperature accelerates bio-mediated carbon remineralisation 565 processes, bringing a lower  $\delta^{13}$ C value than CO<sub>2</sub> coming from water-atmosphere equilibration 566 and rock-leaching CO<sub>2</sub> (Polsenaere et al., 2013 and references therein). Consequently, 567 phytoplankton  $\delta^{13}$ C decreases as phytoplankton uses remineralised CO<sub>2</sub> and thus as water 568 temperature increases. Phytoplankton  $\delta^{15}N$  depends on N-nutrient origin and availability 569 570 (Savoye et al., 2003 and references therein). Especially, it increases with nutrient concentration 571 decrease (Sigman et al., 2009) as reported for the Rance River (Tab. 2). Water conductivity 572 could be considered as a proxy of water mass and thus of nitrate origin. This may explain the relationship between phytoplankton  $\delta^{15}$ N and water conductivity in the Orb River (Tab. 2). 573

- In the studied rivers, phytoplankton  $\delta^{13}$ C,  $\delta^{15}$ N and C/N ratio ranged between -34.9  $\pm$  0.4 and -
- $23.8 \pm 0.6$  %,  $4.3 \pm 0.8$  and  $13.2 \pm 1.8$  %, and  $4.8 \pm 0.9$  and  $10 \pm 0.9$  mol/mol, respectively.
- This is similar to values reported for the Loire River, another French river (-30.6  $\leq \delta^{13}$ C  $\leq$  -25.0
- 577 %;  $3.0 \le \delta^{15}$ N  $\le 10.4$  %; C/N =  $7.2 \pm 0.6$  mol/mol: Ferchiche et al., 2024), but narrower ranges
- can be found in the literature. In the Sava River (Eastern Europe), phytoplankton signature was
- $-30.4 \pm 2.1$  %,  $5.0 \pm 1.5$  % and  $6.5 \pm 1.5$  mol/mol for  $\delta^{13}$ C,  $\delta^{15}$ N and C/N ratio, respectively
- (Ogrinc et al., 2008), similar to that of Indian ( $\delta^{13}$ C = -30.6 ± 1.7 ‰,  $\delta^{15}$ N = 7.0 ± 2.3 ‰;
- Gawade et al., 2018) and Texan ( $\delta^{13}$ C = -31,4 %; Lebreton et al., 2016) rivers. Lower  $\delta^{13}$ C
- values ( $\leq$  -32 %) were also found (Finlay et al., 2010; Hellings et al., 1999; Sato et al., 2006;
- Savoye et al., 2012). However, values of elemental and isotopic ratios for riverine
- phytoplankton are scarce in the literature. Indeed, it is not easy to estimate the phytoplankton
- signature since it cannot be separated from other particles. Thus, literature estimates may not
- be perfectly representative of the variability of phytoplankton isotopic signatures.

# 4.2. Watershed characteristics drive spatial dynamics of POM composition

- At the annual scale, we observed deep variations between studied rivers regarding the mean
- POC proportion of the different sources ( $5 \le \text{phytoplankton} \le 80 \%$ ;  $17 \le \text{labile terrestrial POC}$
- $\leq$  95 %;  $0 \leq$  refractory terrestrial POC  $\leq$  39 %).

587

- Interestingly, phytoplankton proportions were highly correlated to the proportion of agriculture
- surface areas and conductivity and in a less extent to river nitrate concentration (Fig. 7). Such
- relationship between agriculture surface and phytoplankton is well-known, as agricultural
- activities increase nutrient inputs to river bodies (Khan and Mohammad, 2014), leading to better
- conditions for phytoplankton growth (Dodds and Smith, 2016; Minaudo et al., 2015).
- Also interestingly, the proportions of labile terrestrial matter were positively correlated to soil
- organic carbon content, soil erosion, the acidic soil coverage and sandy material coverage (Fig.
- 7, Fig. A6), indicating a strong relationship between terrestrial matter in rivers and soil nature
- with undecomposed and fresh detrital matter (McCorkle et al., 2016). They are also negatively
- correlated to phytoplankton proportions and their related environmental parameters. Rivers
- which POM is dominated by labile terrestrial POM (mainly rivers of type I) flow through
- catchments dominated by sandy material in podzol (an acidic soil type). This kind of soil is
- subject to soil erosion and releases large amounts of coloured dissolved organic carbon,
- favouring the input of terrestrial material (soil erosion) and disfavouring phytoplankton growth
- in the river water because of the turbidity due to the dissolved organic carbon (Canton et al.,
- 2012; Polsenaere et al., 2013).
- The proportions of refractory terrestrial matter are correlated to the catchment slope and
- negatively correlated to useful water reserve (Fig. 7, Fig. A6). Rivers for which POM is partly

composed of refractory terrestrial POM (most of the rivers of type II) flow through catchments of mountainous surfaces, which are associated with shallower, poorer topsoil and more outcropping bedrock (so a reduced useful water reserve of the soil). It favours more reactive and abrupt transfer of water to the river, leading to enhanced episodes of sediment resuspension, as well as permitting a rock-derived POM weathering (Copard et al., 2018; Higueras et al., 2014; Yaalon, 1997).

629630

633634

# 4.3. Temporal dynamics of POM composition and river-dynamics typology

If average quantitative difference between rivers can be input to differences in the catchment characteristics (see section 4.2), seasonally, phytoplankton likely appears during spring and summer in favourable conditions, related to low discharge, high-temperature conditions and enough nutrients to support its growth, while in winter, high turbidity and low-temperature conditions limit its presence (Turner et al., 2022). Terrestrial material likely appears during winter conditions, related to floods that transport great amounts of terrestrial material (Dalzell et al., 2007). Such a seasonal dichotomy between phytoplankton and terrestrial POM was clearly visible for most of the studied rivers (Fig. 4), especially for type-IV and type-III rivers, but even for some of those highly dominated by the labile terrestrial POM (e.g., Milieu and Tagon Rivers; Type-I rivers). This was illustrated by the relationships between phytoplankton POM and chlorophyll a concentration and/or temperature (as proxies of favourable conditions for phytoplankton production) on the one hand, and between labile terrestrial POM and river flow and/or SPM concentration on the other hand (Fig. 6 and A5). This dichotomy in POM composition was also reported in other similar studies (e.g., Kelso and Baker, 2020; Lu et al., 2016). In rivers where refractory terrestrial POM was present in addition to the labile one (type-II rivers), both terrestrial sources were linked to river flows and SPM concentrations. More precisely, it was interesting to see that the refractory terrestrial POM was more related to SPM concentration than river flow and inversely for the labile terrestrial POM. This indicates that labile and refractory terrestrial POM were preferentially associated with direct river input and sediment resuspension, respectively. The origin of the refractory terrestrial POM may be fossil/bedrock/petrogenic OM (e.g., Copard et al., 2022; Hilton et al., 2010; Sun et al., 2021) brought by river flow (in quantity undetectable in the bulk POM using our tools), especially in Type-II rivers which watersheds are characterized by high slopes (Fig. 7). This POM can be accumulated in downstream sediments and be resuspended (in quantity calculable in the bulk POM using our tools). Refractory terrestrial POM may also come from labile terrestrial POM brought by the river flow and then accumulated and reworked/decayed until refractory POM in the sediment (e.g., Etcheber et al., 2007; Savoye et al., 2012), which can be resuspended.

Sewage POM was detected in two of the studied rivers, but with different associated temporal

dynamics. In the Têt River, because the former WWTP was dysfunctional, a new one replaced

it in late 2007 (https://www.assainissement.developpement-durable.gouv.fr/pages/data/fiche-060966136002, last visit 10/09/24). This explains the shift in sewage POM between the two 648 649 studied periods (2006-2007 versus 2008-2010 without anthropogenic POM). In the Orb River, 650 sewage POM was detected throughout the studied periods. The WWTP is located only a few 651 kilometres upstream of the sampling site and is large enough (220,000 inhabitant equivalent) compared to the river flow (annual mean: 23m<sup>3</sup>/s) to make the sewage POM detectable in the 652 bulk POM using  $\delta^{15}$ N. Such a result is quite common for urban rivers (e.g., Kelso and Baker, 653 654 2020).

### 5. Synthesis, originality of the study and perspectives

674675

The present study provides a comprehensive assessment of POM composition and its spatial and seasonal variability in temperate rivers. By including twenty-three rivers spanning a wide range of environmental conditions under a temperate climate, a river-dynamics typology is proposed based on POM composition and its temporal patterns. In type-I rivers, POM is dominated by labile terrestrial material throughout the year. This material is mainly associated with suspended particulate matter. Phytoplankton makes a slight contribution, especially during summer. Type-II rivers are characterised by the presence of both labile and refractory terrestrial material, along with phytoplankton. The variability between these sources over time is high, but seasonality is not always evident, although phytoplankton and terrestrial POM can dominate the POM composition during summer and winter, respectively. Nonetheless, if both terrestrial sources are primarily linked to river flow and SPM, a better coupling of refractory terrestrial POM with SPM indicates that this material is probably stored in sediments and resuspended, whatever its origin (soil, litter, petrogenic POM). In type-III rivers, POM consists of phytoplankton and labile terrestrial material. The seasonality of POM composition is not very pronounced, though the contribution of labile terrestrial POM is closely related to river flow. Type III is an intermediate between type I and type IV. In type-IV rivers, POM is also composed of phytoplankton and labile terrestrial material, but the seasonality is highly marked, with a clear shift from high phytoplankton contribution in summer to high terrestrial contribution in winter. Labile terrestrial POM remains closely associated with river flow. Beyond this typology, the main differences in POM composition between the studied rivers are related to catchment inherent properties. The contribution of phytoplankton is correlated with the proportion of agricultural coverage, while the contribution of labile terrestrial POM is linked to leached OM-rich thick soil features and the refractory terrestrial POM to thin OM-poor soils with high rock-derived features.

The originality of the present study lies firstly in its approach. Even if C and N stable isotopes

have been used for decades to investigate POM origins within river waters, the quantification

of POM composition (i.e., the relative proportion of each source composing the POM) using

mixing models, especially Bayesian mixing models, is not so common. Most previous studies

either use literature data for phytoplankton isotopic signature (e.g., Zhang et al., 2021) or use lake or autochthonous POM as a proxy of phytoplankton (e.g., Kelso and Baker, 2020). Also, most of these studies use direct measurements of soil or plants to assess the isotopic signature of terrestrial POM, although this material may rework within the water column or sediment, changing its elemental and isotopic values (e.g., Savoye et al., 2012). These approaches do not consider the temporal variability of phytoplankton and terrestrial material isotopic signatures. In the present study, we used the approach developed by Savoye et al. (2012) in an estuary, Liénart et al. (2017) in coastal systems, and Ferchiche et al. (2024, 2025) in a river to assess elemental and isotopic signatures from subsets of bulk POM and, when needed, empirical equations. This approach has the advantage of 1) using signatures dedicated to the sampling area and 2) taking into account the potential variability of these signatures over time, i.e., depending on environmental conditions for phytoplankton growth and its decay for phytoplankton and terrestrial POM. Especially, we discriminated labile from refractory terrestrial POM in some rivers, as Savoye et al. (2012) did in an estuary. Another great originality of the present study lies in the multi-system approach: studying 23 rivers in a single study allowed the detection of four types of river functioning regarding the POM composition and its temporal dynamics, which has not been performed before. It also highlights the great influence of land use (agriculture) and characteristics (erosion, organic carbon content, type of soil) on the POM composition of rivers. At last, the multi-parameter use of  $\delta^{13}$ C,  $\delta^{15}$ N, and C/N ratio allowed either to perform mixing models with up to four end-members or to study POC and PN composition separately. It showed that POC and PN display very similar compositions and dynamics in rivers. Overall, this study, which focuses on the River-Estuary Interface, brings meaningful information for the comprehension of C and N cycles along the LOAC and especially the behaviour, dynamics and drivers of POM that leaves the river and enters the estuary.

From a methodological perspective, such a study could be strengthened by the use of non-exchangeable  $\delta^2H$  as an additional tool to even better distinguish and quantify more sources in mixing models. This tool has been recently shown to be powerful for such purposes (Ferchiche et al., 2025). From a fundamental perspective, aggregating more datasets from other temperate rivers would allow testing the robustness of this typology and probably detecting additional types, but also datasets from polar and tropical rivers to perform an even more comprehensive study at a global climate scale. Clearly, the approach developed in the present study is transferable to other regions of the planet and used at broader spatial scales. In addition, a similar study dedicated to the estuarine systems would even increase our comprehensive understanding of the origin and fate of POM along the Land-Ocean Aquatic Continuum by complementing the present study dedicated to the River-Estuary Interface and those of Liénart et al. (2017, 2018) dedicated to the coastal systems.

686

#### Appendix a typology of river functioning: **Toward** comprehensive study of the particulate organic matter composition at the multi-river scale Florian Ferchiche<sup>1</sup>, Camilla Liénart<sup>1</sup>, Karine Charlier<sup>1</sup>, Jonathan Deborde<sup>2,3</sup>, Mélanie Giraud<sup>4</sup>, Philippe Kerhervé<sup>5</sup>, Pierre Polsenaere<sup>1,3</sup>, Nicolas Savoye<sup>1\*</sup> <sup>1</sup> Univ. Bordeaux, CNRS, EPHE, Bordeaux INP, UMR 5805 EPOC, F-33600 Pessac, France <sup>2</sup> Univ. Pau & Pays Adour, CNRS, E2S UPPA – MIRA, UMR 5254 IPREM, F-64000 Pau, F-64600 Anglet, France <sup>3</sup> Ifremer, COAST, F-17390 La Tremblade, France <sup>4</sup> MNHN, Station Marine de Dinard, F-35800 Dinard, France <sup>5</sup> Univ. Perpignan, CNRS, UMR 5110 CEFREM, F-66860 Perpignan, France \*Correspondence to: Nicolas Savoye (nicolas.savoye@u-bordeaux.fr) (+33)556223916 Université de Bordeaux - UMR 5805 EPOC - Station marine d'Arcachon - 2 rue du Professeur Jolyet - 33120 Arcachon \*Corresponding author nicolas.savoye@u-bordeaux.fr Station marine - 2 rue du Professeur Jolyet 33120 ARCACHON Keywords: River-Estuary Interface; particulate organic matter; isotopes; multi-ecosystem study

Matter; POC or PN = Particulate Organic Carbon or Nitrogen; Chl a = Chlorophyll a; Phaeo = Phaeopigments. database; \*\* means that the data were retrieved from the Météo France database). n = number of sampling dates; SPM = Suspended Particulate Table A1 Summary of data availability and origin (X means that the data were available; \* means that the data were retrieved from the Naïades

| River              | Dates              | Sam                      | Samplings | I atitude | I opartude | 213            | 815NI          | CN    | S S        | POC     | PN | 보     P | haeo ten                | Water      |            | ond              | u P        | Paramet    | Parameters                            | Š.              | NO. BO 3-                          | NO. BO 3. Flow                       | NO - DO 3- Flow Rainfall                      | NO - DO 3- Flow Rainfall                                           | NO - DO 3- Flow Rainfall                                | NO DO 3. Flow Rainfall tempera                                                  |
|--------------------|--------------------|--------------------------|-----------|-----------|------------|----------------|----------------|-------|------------|---------|----|---------|-------------------------|------------|------------|------------------|------------|------------|---------------------------------------|-----------------|------------------------------------|--------------------------------------|-----------------------------------------------|--------------------------------------------------------------------|---------------------------------------------------------|---------------------------------------------------------------------------------|
| River              | Dates              | Periodicity              | п         | Latitude  | Longitude  | $\delta^{13}C$ | $\delta^{15}N$ | ratio | SPM        | SPM POC | PN | Жla Р   | Chl a Phaeo temperat pH | nperat     |            | Condu<br>ctivity |            | + 4        | Condu NH <sub>4</sub> NO <sub>2</sub> | NO <sub>3</sub> | NO <sub>3</sub> PO <sub>4</sub> 3- | NO <sub>3</sub> PO <sub>4</sub> Flow | NO <sub>3</sub> PO <sub>4</sub> Flow Rainfall | NO <sub>3</sub> PO <sub>4</sub> <sup>3</sup> Flow Rainfall tempera | NO <sub>3</sub> PO <sub>4</sub> Flow Rainfall tempera d | NO <sub>3</sub> PO <sub>4</sub> <sup>3</sup> Flow Rainfall tempera direction in |
| Seine              | 06/2014 to 06/2015 | monthly                  | 13        | 49.306667 | 1.242500   | ×              | ×              | ×     | <b>X</b> * | ×       | ×  | ×       | ×                       | X*         | X*         | <b>X</b> *       |            | <b>X</b> * | X* X*                                 |                 | <b>X</b> *                         | X* X*                                | X* X* X*                                      | X* X* X* X                                                         | X* X* X* X X**                                          | X* X* X* X X** X**                                                              |
| Orne               | 06/2014 to 06/2015 | monthly                  | 13        | 49.179722 | -0.349167  | ×              | ×              | ×     | <b>X</b> * | ×       | ×  | ×       | ×                       | <b>X</b> * | *X         | *X               |            | <b>X</b> * | X* X*                                 |                 | <b>X</b> *                         | X* X*                                | X* X* X*                                      | X* X* X* X                                                         | X* X* X* X X**                                          | X* X* X* X X** X**                                                              |
| Rance              | 06/2014 to 05/2015 | monthly                  | 12        | 48.491667 | -2.001389  | ×              | ×              | ×     | <b>X</b> * | ×       | ×  | ×       | ×                       | *X         | X*         | X*               |            | *<br>X*    |                                       | <b>X</b> *      | X* X*                              | X* X* X*                             | X* X* X* X*                                   | X* X* X* X* X                                                      | X* X* X* X* X X**                                       | X* X* X* X* X X** X**                                                           |
| Elorn              | 01/2014 to 06/2015 | monthly                  | 17        | 48.450556 | -4.248333  | ×              | ×              | ×     | <b>X</b> * | ×       | ×  | ×       | ×                       | X*         | <b>X</b> * |                  | <b>X</b> * | X* X*      |                                       | <b>X</b> *      | X* X*                              | X* X* X*                             | X* X* X* X*                                   | X* X* X* X                                                         | X* X* X* X* X X**                                       | X* X* X* X* X X** X**                                                           |
| Aulne              | 01/2014 to 06/2015 | monthly                  | 17        | 48.212778 | -4.094444  | ×              | ×              | ×     | <b>X</b> * | ×       | ×  | ×       | ×                       | *X         | *X         |                  | <b>X</b> * | X* X*      |                                       | <b>X</b> *      | X* X*                              | X* X* X*                             | X* X* X* X*                                   | X* X* X* X                                                         | X* X* X* X* X X**                                       | X* X* X* X* X X** X**                                                           |
| Loire              | 10/2009 to 07/2012 | bi-monthly               | 67        | 47.392095 | -0.860351  | ×              | ×              | ×     | <b>X</b> * | ×       | ×  | ×       | ×                       | ×          | ×          |                  | ×          | x          |                                       | ×               | ×                                  | x<br>x<br>x                          | x                                             | x x x x x                                                          | X X X X X X**                                           | X X X X X X** X**                                                               |
| Sèvre<br>niortaise | 03/2014 to 03/2015 | monthly                  | 13        | 46.315348 | -1.003891  | ×              | ×              | ×     | <b>X</b> * | ×       | ×  | ×       | ×                       | X*         |            | *X               | <b>X</b> * |            | <b>X</b> *                            | X* X*           | X* X* X*                           | X* X* X* X*                          | X* X* X* X* X*                                | X* X* X* X* X                                                      | X* X* X* X* X* X* X*                                    | X* X* X* X* X* X* X*                                                            |
| Charente           | 03/2014 to 03/2015 | monthly                  | 13        | 45.868056 | -0.713056  | ×              | ×              | ×     | <b>X</b> * | ×       | ×  | ×       | ×                       | <b>X</b> * |            | <b>X</b> *       | <b>X</b> * |            | <b>X</b> *                            | X* X*           | X* X* X*                           | X* X* X*                             | X* X* X* X*                                   | X* X* X* X* X                                                      | X* X* X* X* X* X**                                      | X* X* X* X* X* X** X**                                                          |
| Seudre             | 03/2014 to 09/2015 | monthly                  | 15        | 45.674027 | -0.933123  | ×              | ×              | ×     | <b>X</b> * | ×       | ×  | ×       | ×                       | <b>X</b> * |            | <b>X</b> *       | <b>X</b> * |            | <b>X</b> *                            | X* X*           | X* X* X*                           | X* X* X* X*                          | X* X* X* X* X*                                | X* X* X* X* X                                                      | X* X* X* X* X* X X**                                    | X* X* X* X* X* X X** X**                                                        |
| Porge              | 01/2008 to 02/2009 | monthly                  | 14        | 44.789868 | -1.161181  | ×              |                | ×     | ×          | ×       |    | ×       | ×                       | ×          |            |                  | ×          | ×          | ×                                     | X X*            |                                    |                                      | X*                                            | X* X                                                               | X* X X**                                                | X* X X** X**                                                                    |
| Renet              | 02/2008 to 02/2009 | bi-monthly               | 23        | 44.714466 | -1.044013  | ×              |                | ×     | ×          | ×       |    | ×       | ×                       | ×          |            |                  | ×          | ×          | ×                                     | X X*            |                                    |                                      | X*                                            | X* X                                                               | X* X X**                                                | X* X X** X**                                                                    |
| Milieu             | 02/2008 to 02/2009 | monthly                  | 13        | 44.697326 | -1.022532  | ×              |                | ×     | ×          | ×       |    | ×       | ×                       | ×          |            |                  | ×          | ×          | ×                                     | X X*            |                                    |                                      | X*                                            | X* X                                                               | X* X X**                                                | X* X X** X**                                                                    |
| Cirès              | 02/2008 to 02/2009 | monthly                  | 13        | 44.759820 | -1.110657  | ×              |                | ×     | ×          | ×       |    | ×       | ×                       | ×          |            |                  | ×          | ×          | ×                                     | X X*            |                                    |                                      | X*                                            | X* X                                                               | X* X X**                                                | X* X X** X**                                                                    |
| Lanton             | 02/2008 to 02/2009 | monthly                  | 13        | 44.700283 | -1.024385  | ×              |                | ×     | ×          | ×       |    | ×       | ×                       | ×          |            |                  | ×          | ×          | ×                                     | X X*            |                                    |                                      | X*                                            | X* X                                                               | X* X X**                                                | X* X X** X**                                                                    |
| Tagon              | 02/2008 to 02/2009 | bi-monthly               | 26        | 44.659049 | -0.989050  | ×              |                | ×     | ×          | ×       |    | ×       | ×                       | ×          |            |                  | ×          | ×          | ×                                     | X X*            |                                    |                                      | $X^*$                                         | Х* Х                                                               | X* X X**                                                | X* X X** X**                                                                    |
| Landes             | 02/2008 to 02/2009 | monthly                  | 12        | 44.616912 | -1.109066  | ×              |                | ×     | ×          | ×       |    | ×       | ×                       | ×          |            |                  | ×          | ×          | ×                                     | X X*            |                                    |                                      | X*                                            | X* X                                                               | X* X X**                                                | X* X X** X**                                                                    |
| Leyre              | 02/2008 to 02/2015 | bi-monthly<br>or monthly | 59        | 44.626389 | -0.996111  | ×              | ×              | ×     | <b>X</b> * | ×       | ×  | ×       | ×                       | ×          |            | X*               | <b>X</b> * |            | <b>X</b> *                            | X* X*           | X* X* X*                           | X* X* X* X*                          | X* X* X* X* X*                                | X* X* X* X* X* X                                                   | X* X* X* X* X* X* X                                     | X* X* X* X* X* X* X** X**                                                       |
| Adour              | 04/2013 to 05/2018 | monthly                  | 24        | 43.498880 | -1.294899  | ×              | ×              | ×     | ×          | ×       | ×  | ×       | ×                       | ×          |            |                  |            | ×          | X X                                   |                 | ×                                  | ×                                    | x x                                           | x x x                                                              | X X X X**                                               | X X X X** X**                                                                   |
| Têt                | 01/2006 to 05/2010 | monthly                  | 52        | 42.713704 | 2.993488   | ×              | ×              | ×     | ×          | ×       | ×  |         |                         | ×          |            | ×                | ×          |            | ×                                     | X X*            | X X* X*                            | X X* X* X*                           | X X* X* X* X*                                 | X X* X* X* X* X                                                    | X X* X* X* X* X* X                                      | X X* X* X* X* X X** X**                                                         |
| Aude               | 01/2006 to 05/2010 | monthly                  | 52        | 43.244281 | 3.152733   | ×              | ×              | ×     | ×          | ×       | ×  |         |                         | ×          |            | ×                | ×          |            | ×                                     | x<br>x*         | X X* X*                            | X X* X* X*                           | X X* X* X* X*                                 | X X* X* X* X* X                                                    | X X* X* X* X* X X**                                     | X X* X* X* X* X X** X**                                                         |
| Orb                | 01/2006 to 05/2010 | monthly                  | 52        | 43.285004 | 3.281278   | ×              | ×              | ×     | ×          | ×       | ×  |         |                         | ×          |            | ×                | ×          |            | ×                                     | X X*            | X X* X*                            | X X* X* X*                           | X X* X* X* X*                                 | X X* X* X* X* X                                                    | X X* X* X* X* X X**                                     | X X* X* X* X* X X**                                                             |
| Hérault            | 01/2006 to 05/2010 | monthly                  | 52        | 43.359415 | 3.435398   | ×              | ×              | ×     | ×          | ×       | ×  |         |                         | ×          |            | ×                | X          |            | ×                                     | X X*            | X X* X*                            | X X* X* X*                           | X X* X* X* X*                                 | X X* X* X* X* X                                                    | X X* X* X* X* X X**                                     | X X* X* X* X* X X** X**                                                         |
| Rhône              | 12/2003 to 01/2011 | monthly                  | 105       | 43.678724 | 4.621188   | ×              | ×              | ×     | ×          | ×       | ×  | X       | X                       | ×          |            | ×                | x          |            | x                                     | X X*            | X X* X*                            | X X* X* X*                           | X X* X* X* X*                                 | X X* X* X* X* X                                                    | X X* X* X* X* X X**                                     | X X* X* X* X* X X** X**                                                         |
|                    |                    |                          |           |           |            |                |                |       |            |         |    |         |                         |            |            |                  |            |            |                                       |                 |                                    |                                      |                                               |                                                                    |                                                         |                                                                                 |

informative, not available, or auto-correlated (see section 2.5)). Table A2 Summary of parameters kept (informative) to perform the local seasonal RDAs, opposite to those not considered (because non-

Similarities

Similarities

10 cut off levels

3 sources

12 dendrograms

10 dendrograms

12 dendrograms

 $\underline{Figure\ A1}\ Diagram\ detailing\ the\ regionalisation\ method,\ adapted\ from\ Souissi\ et\ al.\ (2000).$ 

river flow (left axis; black dotted line) and temperature (right axis; blue line) (fourth column). and δ<sup>15</sup>N (right axis; blue line) (first column); C/N (left axis; black dotted line) and POC/chl a(right axis; blue line) ratios (second column); SPM(left axis; black dotted line), POC (right axis; blue line) and chlorophyll a (right axis; blue dotted line) concentrations (third column) and Figure A2 Temporal variations of matter characteristics for representative rivers along the studied periods for  $\delta^{13}$ C (left axis; black dotted line)

Figure A3  $\delta^{13}$ C,  $\delta^{15}$ N, N/C and/or C/N values of bulk POM (black crosses) and sources. The latter are presented as closed circles (average) and bars (standard deviation) when the signatures

were constant over time and by colored area when at least one of the proxies was variable over time (see Table 2). This colored area corresponds to the dispersion of the values, including their uncertainties.

<u>Figure A4</u> Temporal dynamic (rectangle graphs) and (inter-)annual mean (pie charts) of PN source proportions. Sources are phytoplankton (green) and labile terrestrial material (brown).

Figure A5 Redundancy analyses (correlation circles) of rivers standing for each type of river. Black arrows represent explained variables (concentration of POC or PN sources) and red arrows represent explaining variables (environmental variables). River types are recalled (Roman numerals). LT<sub>POC or PN</sub> = Labile terrestrial POC or PN; RT<sub>POC</sub> = Refractory terrestrial POC;  $\varphi_{POC \text{ or PN}}$  = Phytoplankton POC or PN; Anth. POC = Anthropogenic POM; SPM = Suspended particulate matter; Chl a = chlorophyll a; Phaeo. = phaeopigments; M. wind = meridional wind; Z. wind = zonal wind; R. flow = river flow; Temp. = temperature; Irrad. = Irradiance; NH<sub>4</sub><sup>+</sup> = ammonium; NO<sub>3</sub><sup>-</sup> = nitrate; PO<sub>4</sub><sup>3-</sup> = phosphates; Adj. R<sup>2</sup> = adjusted R<sup>2</sup>.

Figure A6 Correlogram of multi-system RDA parameters, including source proportions and accompanying parameters. Descriptions of environmental parameters can be retrieved in section 2.5. Temperature = Water temperature; SPM = Suspended particulate matter; OC in soil = Organic carbon proportions in soil; WWTP population equivalent = sewage treatment capacities; WWTP ...River flow = sewage treatment capacities to average river flow ratio.

<u>Figure A7</u> Typology of river dynamics following a hierarchical cluster analysis on POM source proportions. The percentages of membership for each type attributed to each river are shown. Left panel: considering all sampling years for all rivers. Right panel: considering only one stream of type I fuelling the Bay of Arcachon (Milieu River).

| 834                      |                                                                                                                                                                                                                                                                                                                                                                              |
|--------------------------|------------------------------------------------------------------------------------------------------------------------------------------------------------------------------------------------------------------------------------------------------------------------------------------------------------------------------------------------------------------------------|
| 835                      |                                                                                                                                                                                                                                                                                                                                                                              |
| 836                      | Author contributions                                                                                                                                                                                                                                                                                                                                                         |
| 837<br>838<br>839<br>840 | FF: Formal analysis, Investigation, Visualisation, Writing – original draft. CL: Formal analysis, Conceptualisation, Supervision, Writing – original draft, Writing – review & editing. KC: Investigation. JD: Investigation, Visualisation, Writing – review & editing. MG: Investigation. PK: Investigation. PP: Investigation, Visualisation, Writing – review & editing. |
| 841<br>842<br>843        | NS: Conceptualisation, Formal analysis, Investigation, Methodology, Supervision, Writing – original draft, Writing – review & editing.                                                                                                                                                                                                                                       |
| 844                      | Competing interest                                                                                                                                                                                                                                                                                                                                                           |
| 845<br>846               | The authors declare that they have no known competing financial interests or personal relationships that could have appeared to influence the work reported in this paper.                                                                                                                                                                                                   |
| 847                      |                                                                                                                                                                                                                                                                                                                                                                              |
| 848                      | Data availability                                                                                                                                                                                                                                                                                                                                                            |
| 849<br>850               | All POM and environmental data used in this article are stored in Figshare, accessible for the review process through this private link: <a href="https://figshare.com/s/a7101028e6ab5452c4db">https://figshare.com/s/a7101028e6ab5452c4db</a> .                                                                                                                             |
| 851                      |                                                                                                                                                                                                                                                                                                                                                                              |
| 852                      | References                                                                                                                                                                                                                                                                                                                                                                   |
| 853<br>854<br>855<br>856 | Arellano, A. R., Bianchi, T. S., Osburn, C. L., D'Sa, E. J., Ward, N. D., Oviedo-Vargas, D., Joshi, I. D., Ko, D. S., Shields, M. R., Kurian, G., and Green, J.: Mechanisms of Organic Matter Export in Estuaries with Contrasting Carbon Sources, Journal of Geophysical Research: Biogeosciences, 124, 3168–3188, https://doi.org/10.1029/2018JG004868, 2019.              |
| 857<br>858<br>859<br>860 | Barros, G. V., Martinelli, L. A., Oliveira Novais, T. M., Ometto, J. P. H. B., and Zuppi, G. M.: Stable isotopes of bulk organic matter to trace carbon and nitrogen dynamics in an estuarine ecosystem in Babitonga Bay (Santa Catarina, Brazil), Science of The Total Environment, 408, 2226–2232, https://doi.org/10.1016/j.scitotenv.2010.01.060, 2010.                  |
| 861<br>862<br>863        | Bate, G. C., Whitfield, A. K., Adams, J. B., Huizinga, P., and Wooldridge, T. H.: The importance of the river-estuary interface (REI) zone in estuaries, Water SA, 28, 271–280, https://doi.org/10.4314/wsa.v28i3.4894, 2002.                                                                                                                                                |
| 864<br>865<br>866<br>867 | Bonin, P., Prime, AH., Galeron, MA., Guasco, S., and Rontani, JF.: Enhanced biotic degradation of terrestrial POM in an estuarine salinity gradient: interactive effects of organic matter pools and changes of bacterial communities, Aquatic Microbial Ecology, 83, 147–159, https://doi.org/10.3354/ame01908, 2019.                                                       |

- Borcard, D., Legendre, P., and Gillet, F.: Numerical Ecology with R, Springer, 315 pp., 2011.
- Bouwman, L., Beusen, A., Glibert, P. M., Overbeek, C., Pawlowski, M., Herrera, J., Mulsow,
- S., Yu, R., and Zhou, M.: Mariculture: significant and expanding cause of coastal nutrient
- enrichment, Environ. Res. Lett., 8, 044026, https://doi.org/10.1088/1748-9326/8/4/044026,
- 2013.
- Brett, M. T., Bunn, S. E., Chandra, S., Galloway, A. W. E., Guo, F., Kainz, M. J., Kankaala,
- P., Lau, D. C. P., Moulton, T. P., Power, M. E., Rasmussen, J. B., Taipale, S. J., Thorp, J. H.,
- and Wehr, J. D.: How important are terrestrial organic carbon inputs for secondary production
- in freshwater ecosystems?, Freshwater Biology, 62, 833–853,
- https://doi.org/10.1111/fwb.12909, 2017.
- Canton, M., Anschutz, P., Coynel, A., Polsenaere, P., Auby, I., and Poirier, D.: Nutrient
- export to an Eastern Atlantic coastal zone: first modeling and nitrogen mass balance,
- Biogeochemistry, 107, 361–377, https://doi.org/10.1007/s10533-010-9558-7, 2012.
- Canuel, E. A. and Hardison, A. K.: Sources, Ages, and Alteration of Organic Matter in
- Estuaries, Annual Review of Marine Science, 8, 409–434, https://doi.org/10.1146/annurev-
- marine-122414-034058, 2016.
- Cathalot, C., Rabouille, C., Tisnérat-Laborde, N., Toussaint, F., Kerhervé, P., Buscail, R.,
- Loftis, K., Sun, M.-Y., Tronczynski, J., Azoury, S., Lansard, B., Treignier, C., Pastor, L., and
- Tesi, T.: The fate of river organic carbon in coastal areas: A study in the Rhône River delta
- using multiple isotopic ( $\delta$ 13C,  $\Delta$ 14C) and organic tracers, Geochimica et Cosmochimica
- Acta, 118, 33–55, https://doi.org/10.1016/j.gca.2013.05.001, 2013.
- Chevalier, N., Savoye, N., Dubois, S., Lama, M. L., David, V., Lecroart, P., le Ménach, K.,
- and Budzinski, H.: Precise indices based on *n*-alkane distribution for quantifying sources of
- sedimentary organic matter in coastal systems, Organic Geochemistry, 88, 69–77,
- https://doi.org/10.1016/j.orggeochem.2015.07.006, 2015.
- Copard, Y., Eyrolle, F., Radakovitch, O., Poirel, A., Raimbault, P., Gairoard, S., and Di-
- Giovanni, C.: Badlands as a hot spot of petrogenic contribution to riverine particulate organic
- carbon to the Gulf of Lion (NW Mediterranean Sea), Earth Surface Processes and Landforms,
- 43, 2495–2509, https://doi.org/10.1002/esp.4409, 2018.
- Copard, Y., Eyrolle, F., Grosbois, C., Lepage, H., Ducros, L., Morereau, A., Bodereau, N.,
- Cossonnet, C., and Desmet, M.: The unravelling of radiocarbon composition of organic
- carbon in river sediments to document past anthropogenic impacts on river systems, Science
- of The Total Environment, 806, 150890, https://doi.org/10.1016/j.scitotenv.2021.150890,
- 2022.
- Dagg, M., Benner, R., Lohrenz, S., and Lawrence, D.: Transformation of dissolved and
- particulate materials on continental shelves influenced by large rivers: plume processes,
- Continental Shelf Research, 24, 833–858, https://doi.org/10.1016/j.csr.2004.02.003, 2004.
- Dalzell, B. J., Filley, T. R., and Harbor, J. M.: The role of hydrology in annual organic carbon
- loads and terrestrial organic matter export from a midwestern agricultural watershed,
- Geochimica et Cosmochimica Acta, 71, 1448–1462,
- https://doi.org/10.1016/j.gca.2006.12.009, 2007.

- David, V., Sautour, B., Chardy, P., and Leconte, M.: Long-term changes of the zooplankton
- variability in a turbid environment: The Gironde estuary (France), Estuarine, Coastal and
- Shelf Science, 64, 171–184, https://doi.org/10.1016/j.ecss.2005.01.014, 2005.
- Deborde, J.: Dynamique des sels nutritifs et de la matière organique dans le système
- fluvioestuarien de l'Adour/Golfe de Gascogne, Université de Pau et des Pays de l'adour,
- 2019.
- Dodds, W. K. and Smith, V. H.: Nitrogen, phosphorus, and eutrophication in streams, Inland
- Waters, 6, 155–164, https://doi.org/10.5268/IW-6.2.909, 2016.
- Dubois, S., Savoye, N., Grémare, A., Plus, M., Charlier, K., Beltoise, A., and Blanchet, H.:
- Origin and composition of sediment organic matter in a coastal semi-enclosed ecosystem: An
- elemental and isotopic study at the ecosystem space scale, Journal of Marine Systems, 94, 64
- 73, https://doi.org/10.1016/j.jmarsys.2011.10.009, 2012.
- Dunn†, J. C.: Well-Separated Clusters and Optimal Fuzzy Partitions, Journal of Cybernetics,
- 4, 95–104, https://doi.org/10.1080/01969727408546059, 1974.
- Dürr, H. H., Laruelle, G. G., van Kempen, C. M., Slomp, C. P., Meybeck, M., and
- Middelkoop, H.: Worldwide Typology of Nearshore Coastal Systems: Defining the Estuarine
- Filter of River Inputs to the Oceans, Estuaries and Coasts, 34, 441–458,
- https://doi.org/10.1007/s12237-011-9381-y, 2011.
- Etcheber, H., Taillez, A., Abril, G., Garnier, J., Servais, P., Moatar, F., and Commarieu, M.-
- 928 V.: Particulate organic carbon in the estuarine turbidity maxima of the Gironde, Loire and
- Seine estuaries: origin and lability, Hydrobiologia, 588, 245–259,
- https://doi.org/10.1007/s10750-007-0667-9, 2007.
- Falkowski, P. G., Barber, R. T., and Smetacek, V.: Biogeochemical Controls and Feedbacks
- on Ocean Primary Production, Science, 281, 200–206,
- https://doi.org/10.1126/science.281.5374.200, 1998.
- Ferchiche, F., Liénart, C., Charlier, K., Coynel, A., Gorse-Labadie, L., and Savoye, N.:
- Quantifying particulate organic matter: source composition and fluxes at the river-estuary
- interface, Front. Freshw. Sci., 2, https://doi.org/10.3389/ffwsc.2024.1437431, 2024.
- Ferchiche, F., Liénart, C., Savoye, N., and Wassenaar, L. I.: Unlocking the potential of
- hydrogen isotopes (δ2H) in tracing riverine particulate organic matter sources and dynamics,
- Aquat Sci, 87, 5, https://doi.org/10.1007/s00027-024-01127-1, 2025.
- Fernandez, I., Mahieu, N., and Cadisch, G.: Carbon isotopic fractionation during
- decomposition of plant materials of different quality, Global Biogeochemical Cycles, 17,
- https://doi.org/10.1029/2001GB001834, 2003.
- Field, C. B., Behrenfeld, M. J., Randerson, J. T., and Falkowski, P.: Primary Production of the
- Biosphere: Integrating Terrestrial and Oceanic Components, Science, 281, 237–240,
- https://doi.org/10.1126/science.281.5374.237, 1998.
- Finlay, J. C., Doucett, R. R., and McNEELY, C.: Tracing energy flow in stream food webs
- using stable isotopes of hydrogen, Freshwater Biology, 55, 941–951,
- https://doi.org/10.1111/j.1365-2427.2009.02327.x, 2010.

- Fry, B.: 13C/12C fractionation by marine diatoms, Marine Ecology Progress Series, 134,
- 283–294, https://doi.org/10.3354/meps134283, 1996.
- Galeron, M.-A., Radakovitch, O., Charrière, B., Vaultier, F., and Rontani, J.-F.: Autoxidation
- as a major player in the fate of terrestrial particulate organic matter in seawater, Journal of
- Geophysical Research: Biogeosciences, 122, 1203–1215,
- https://doi.org/10.1002/2016JG003708, 2017.
- Gawade, L., Krishna, M. S., Sarma, V. V. S. S., Hemalatha, K. P. J., and Venkateshwara Rao,
- Y.: Spatio-temporal variability in the sources of particulate organic carbon and nitrogen in a
- tropical Godavari estuary, Estuarine, Coastal and Shelf Science, 215, 20–29,
- https://doi.org/10.1016/j.ecss.2018.10.004, 2018.
- Golubkov, M. S., Nikulina, V. N., Tiunov, A. V., and Golubkov, S. M.: Stable C and N
- Isotope Composition of Suspended Particulate Organic Matter in the Neva Estuary: The Role
- of Abiotic Factors, Productivity, and Phytoplankton Taxonomic Composition, Journal of
- Marine Science and Engineering, 8, 959, https://doi.org/10.3390/jmse8120959, 2020.
- Goñi, M. A., Voulgaris, G., and Kim, Y. H.: Composition and fluxes of particulate organic
- matter in a temperate estuary (Winyah Bay, South Carolina, USA) under contrasting physical
- forcings, Estuarine, Coastal and Shelf Science, 85, 273–291,
- https://doi.org/10.1016/j.ecss.2009.08.013, 2009.
- Govan, E. and Parnell, A.: simmr: A Stable Isotope Mixing Model, 2024.
- Grunicke, F., Wagner, A., von Elert, E., Weitere, M., and Berendonk, T.: Riparian detritus vs.
- stream detritus: food quality determines fitness of juveniles of the highly endangered
- freshwater pearl mussels (Margaritifera margaritifera), Hydrobiologia, 850, 729–746,
- https://doi.org/10.1007/s10750-022-05120-3, 2023.
- Harmelin-Vivien, M., Dierking, J., Bănaru, D., Fontaine, M. F., and Arlhac, D.: Seasonal
- variation in stable C and N isotope ratios of the Rhone River inputs to the Mediterranean Sea
- (2004–2005), Biogeochemistry, 100, 139–150, https://doi.org/10.1007/s10533-010-9411-z,
- 2010.
- Hellings, L., Dehairs, F., Tackx, M., Keppens, E., and Baeyens, W.: Origin and fate of
- organic carbon in the freshwater part of the Scheldt Estuary as traced by stable carbon isotope
- composition, Biogeochemistry, 47, 167–186, https://doi.org/10.1023/A:1006143827118,
- 1999.
- Higueras, M., Kerhervé, P., Sanchez-Vidal, A., Calafat, A., Ludwig, W., Verdoit-Jarraya, M.,
- Heussner, S., and Canals, M.: Biogeochemical characterization of the riverine particulate
- organic matter transferred to the NW Mediterranean Sea, Biogeosciences, 11, 157–172,
- https://doi.org/10.5194/bg-11-157-2014, 2014.
- Hilton, R. G., Galy, A., Hovius, N., Horng, M.-J., and Chen, H.: The isotopic composition of
- particulate organic carbon in mountain rivers of Taiwan, Geochimica et Cosmochimica Acta,
- 74, 3164–3181, https://doi.org/10.1016/j.gca.2010.03.004, 2010.
- Hou, P., Eglinton, T. I., Yu, M., Montluçon, D. B., Haghipour, N., Zhang, H., Jin, G., and
- 288 Zhao, M.: Degradation and Aging of Terrestrial Organic Carbon within Estuaries:

- Biogeochemical and Environmental Implications, Environmental Science and Technology,
- 55, 10852–10861, https://doi.org/10.1021/acs.est.1c02742, 2021.
- Hounshell, A. G., Fegley, S. R., Hall, N. S., Osburn, C. L., and Paerl, H. W.: Riverine
- Discharge and Phytoplankton Biomass Control Dissolved and Particulate Organic Matter
- Dynamics over Spatial and Temporal Scales in the Neuse River Estuary, North Carolina,
- Estuaries and Coasts, 45, 96–113, https://doi.org/10/gj64px, 2022.
- INRA: Base de Données Géographique des Sols de France à 1/1 000 000 version 3.2.8.0,
- 10/09/1998, https://doi.org/10.15454/BPN57S, 2025.
- Kang, S., Kim, J.-H., Joe, Y. J., Jang, K., Nam, S.-I., and Shin, K.-H.: Long-term
- environmental changes in the Geum Estuary (South Korea): Implications of river
- impoundments, Marine Pollution Bulletin, 168, 112383, https://doi.org/10/gnpb2t, 2021.
- Ke, Z., Tan, Y., Huang, L., Liu, J., Xiang, C., Zhao, C., and Zhang, J.: Significantly depleted
- 15N in suspended particulate organic matter indicating a strong influence of sewage loading
- in Daya Bay, China, Science of The Total Environment, 650, 759–768,
- https://doi.org/10.1016/j.scitotenv.2018.09.076, 2019.
- Kelso, J. E. and Baker, M. A.: Organic Matter Is a Mixture of Terrestrial, Autochthonous, and
- Wastewater Effluent in an Urban River, Frontiers in Environmental Science, 7,
- https://doi.org/10.3389/fenvs.2019.00202, 2020.
- Kelso, J. E. and Baker, M. A.: Organic matter sources and composition in four watersheds
- with mixed land cover, Hydrobiologia, 849, 2663–2682, https://doi.org/10.1007/s10750-022-
- 04884-y, 2022.
- Kendall, C., Silva, S. R., and Kelly, V. J.: Carbon and nitrogen isotopic compositions of
- particulate organic matter in four large river systems across the United States, Hydrological
- Processes, 15, 1301–1346, https://doi.org/10.1002/hyp.216, 2001.
- Khan, M. N. and Mohammad, F.: Eutrophication: Challenges and Solutions, in:
- Eutrophication: Causes, Consequences and Control: Volume 2, edited by: Ansari, A. A. and
- Gill, S. S., Springer Netherlands, Dordrecht, 1–15, https://doi.org/10.1007/978-94-007-7814-
- 6 1, 2014.
- Lambert, T., Bouillon, S., Darchambeau, F., Morana, C., Roland, F. A. E., Descy, J.-P., and
- Borges, A. V.: Effects of human land use on the terrestrial and aquatic sources of fluvial
- organic matter in a temperate river basin (The Meuse River, Belgium), Biogeochemistry, 136,
- 191–211, https://doi.org/10.1007/s10533-017-0387-9, 2017.
- Le Bas, C.: Carte de la Réserve Utile en eau issue de la Base de Données Géographique des
- Sols de France, https://doi.org/10.15454/JPB9RB, 2025.
- Lebreton, B., Beseres Pollack, J., Blomberg, B., Palmer, T. A., Adams, L., Guillou, G., and
- Montagna, P. A.: Origin, composition and quality of suspended particulate organic matter in
- relation to freshwater inflow in a South Texas estuary, Estuarine, Coastal and Shelf Science,
- 170, 70–82, https://doi.org/10.1016/j.ecss.2015.12.024, 2016.

- Legendre, P., Oksanen, J., and ter Braak, C. J. F.: Testing the significance of canonical axes in
- redundancy analysis, Methods in Ecology and Evolution, 2, 269–277,
- https://doi.org/10.1111/j.2041-210X.2010.00078.x, 2011.
- Lheureux, A., David, V., Del Amo, Y., Soudant, D., Auby, I., Ganthy, F., Blanchet, H.,
- Cordier, M.-A., Costes, L., Ferreira, S., Mornet, L., Nowaczyk, A., Parra, M., D'Amico, F.,
- Gouriou, L., Meteigner, C., Oger-Jeanneret, H., Rigouin, L., Rumebe, M., Tournaire, M.-P.,
- Trut, F., Trut, G., and Savoye, N.: Bi-decadal changes in nutrient concentrations and ratios in
- marine coastal ecosystems: The case of the Arcachon bay, France, Progress in Oceanography,
- 201, 102740, https://doi.org/10.1016/j.pocean.2022.102740, 2022.
- Liénart, C., Susperregui, N., Rouaud, V., Cavalheiro, J., David, V., Del Amo, Y., Duran, R.,
- Lauga, B., Monperrus, M., Pigot, T., Bichon, S., Charlier, K., and Savoye, N.: Dynamics of
- particulate organic matter in a coastal system characterized by the occurrence of marine
- mucilage A stable isotope study, Journal of Sea Research, 116, 12–22,
- https://doi.org/10.1016/j.seares.2016.08.001, 2016.
- Liénart, C., Savoye, N., Bozec, Y., Breton, E., Conan, P., David, V., Feunteun, E., Grangeré,
- 1042 K., Kerhervé, P., Lebreton, B., Lefebvre, S., L'Helguen, S., Mousseau, L., Raimbault, P.,
- Richard, P., Riera, P., Sauriau, P.-G., Schaal, G., Aubert, F., Aubin, S., Bichon, S., Boinet, C.,
- Bourasseau, L., Bréret, M., Caparros, J., Cariou, T., Charlier, K., Claquin, P., Cornille, V.,
- Corre, A.-M., Costes, L., Crispi, O., Crouvoisier, M., Czamanski, M., Del Amo, Y.,
- Derriennic, H., Dindinaud, F., Durozier, M., Hanquiez, V., Nowaczyk, A., Devesa, J.,
- Ferreira, S., Fornier, M., Garcia, F., Garcia, N., Geslin, S., Grossteffan, E., Gueux, A.,
- Guillaudeau, J., Guillou, G., Joly, O., Lachaussée, N., Lafont, M., Lamoureux, J., Lecuyer, E.,
- Lehodey, J.-P., Lemeille, D., Leroux, C., Macé, E., Maria, E., Pineau, P., Petit, F., Pujo-Pay,
- 1050 M., Rimelin-Maury, P., and Sultan, E.: Dynamics of particulate organic matter composition in
- coastal systems: A spatio-temporal study at multi-systems scale, Progress in Oceanography,
- 156, 221–239, https://doi.org/10/gb27ss, 2017.
- Liénart, C., Savoye, N., David, V., Ramond, P., Rodriguez Tress, P., Hanquiez, V., Marieu,
- 1054 V., Aubert, F., Aubin, S., Bichon, S., Boinet, C., Bourasseau, L., Bozec, Y., Bréret, M.,
- Breton, E., Caparros, J., Cariou, T., Claquin, P., Conan, P., Corre, A.-M., Costes, L.,
- Crouvoisier, M., Del Amo, Y., Derriennic, H., Dindinaud, F., Duran, R., Durozier, M.,
- Devesa, J., Ferreira, S., Feunteun, E., Garcia, N., Geslin, S., Grossteffan, E., Gueux, A.,
- Guillaudeau, J., Guillou, G., Jolly, O., Lachaussée, N., Lafont, M., Lagadec, V., Lamoureux,
- J., Lauga, B., Lebreton, B., Lecuyer, E., Lehodey, J.-P., Leroux, C., L'Helguen, S., Macé, E.,
- Maria, E., Mousseau, L., Nowaczyk, A., Pineau, P., Petit, F., Pujo-Pay, M., Raimbault, P.,
- Rimmelin-Maury, P., Rouaud, V., Sauriau, P.-G., Sultan, E., and Susperregui, N.: Dynamics
- of particulate organic matter composition in coastal systems: Forcing of spatio-temporal
- variability at multi-systems scale, Progress in Oceanography, 162, 271–289,
- https://doi.org/10.1016/j.pocean.2018.02.026, 2018.
- Liénart, C., Savoye, N., Conan, P., David, V., Barbier, P., Bichon, S., Charlier, K., Costes, L.,
- Derriennic, H., Ferreira, S., Gueux, A., Hubas, C., Maria, E., and Meziane, T.: Relationship
- between bacterial compartment and particulate organic matter (POM) in coastal systems: An
- assessment using fatty acids and stable isotopes, Estuarine, Coastal and Shelf Science, 239,
- 106720, https://doi.org/10.1016/j.ecss.2020.106720, 2020.

- Lowe, A. T., Galloway, A. W. E., Yeung, J. S., Dethier, M. N., and Duggins, D. O.: Broad
- sampling and diverse biomarkers allow characterization of nearshore particulate organic
- matter, Oikos, 123, 1341–1354, https://doi.org/10.1111/oik.01392, 2014.
- Lu, L., Cheng, H., Pu, X., Wang, J., Cheng, Q., and Liu, X.: Identifying organic matter
- sources using isotopic ratios in a watershed impacted by intensive agricultural activities in
- Northeast China, Agriculture, Ecosystems & Environment, 222, 48–59,
- https://doi.org/10.1016/j.agee.2015.12.033, 2016.
- McCorkle, E. P., Berhe, A. A., Hunsaker, C. T., Johnson, D. W., McFarlane, K. J., Fogel, M.
- 1078 L., and Hart, S. C.: Tracing the source of soil organic matter eroded from temperate forest
- catchments using carbon and nitrogen isotopes, Chemical Geology, 445, 172–184,
- https://doi.org/10.1016/j.chemgeo.2016.04.025, 2016.
- Michener, R. H. and Kaufman, L.: Stable Isotope Ratios as Tracers in Marine Food Webs: An
- Update, 1st ed., edited by: Michener, R. and Lajtha, K., Wiley,
- https://doi.org/10.1002/9780470691854.ch9, 2007.
- Middelburg, J. J. and Herman, P. M. J.: Organic matter processing in tidal estuaries, Marine
- Chemistry, 106, 127–147, https://doi.org/10.1016/j.marchem.2006.02.007, 2007.
- Miller, R. J., Page, H. M., and Brzezinski, M. A.: δ13C and δ15N of particulate organic
- matter in the Santa Barbara Channel: drivers and implications for trophic inference, Marine
- Ecology Progress Series, 474, 53–66, https://doi.org/10.3354/meps10098, 2013.
- Minaudo, C., Meybeck, M., Moatar, F., Gassama, N., and Curie, F.: Eutrophication mitigation
- in rivers: 30 years of trends in spatial and seasonal patterns of biogeochemistry of the Loire
- River (1980–2012), Biogeosciences, 12, 2549–2563, https://doi.org/10.5194/bg-12-2549-
- 2015, 2015.
- Minaudo, C., Moatar, F., Coynel, A., Etcheber, H., Gassama, N., and Curie, F.: Using recent
- high-frequency surveys to reconstitute 35 years of organic carbon variations in a eutrophic
- lowland river, Environ Monit Assess, 188, 41, https://doi.org/10.1007/s10661-015-5054-9,
- 2016.
- Ogrinc, N., Markovics, R., Kanduč, T., Walter, L. M., and Hamilton, S. K.: Sources and
- transport of carbon and nitrogen in the River Sava watershed, a major tributary of the River
- Danube, Applied Geochemistry, 23, 3685–3698,
- https://doi.org/10.1016/j.apgeochem.2008.09.003, 2008.
- O'Leary, M. H.: Carbon isotope fractionation in plants, Phytochemistry, 20, 553–567,
- https://doi.org/10.1016/0031-9422(81)85134-5, 1981.
- Onstad, G. D., Canfield, D. E., Quay, P. D., and Hedges, J. I.: Sources of particulate organic
- matter in rivers from the continental usa: lignin phenol and stable carbon isotope
- compositions, Geochimica et Cosmochimica Acta, 64, 3539–3546,
- https://doi.org/10.1016/S0016-7037(00)00451-8, 2000.
- Parnell, A. C., Phillips, D. L., Bearhop, S., Semmens, B. X., Ward, E. J., Moore, J. W.,
- Jackson, A. L., Grey, J., Kelly, D. J., and Inger, R.: Bayesian stable isotope mixing models,
- Environmetrics, 24, 387–399, https://doi.org/10.1002/env.2221, 2013.

- Perdue, E. M. and Koprivnjak, J.-F.: Using the C/N ratio to estimate terrigenous inputs of
- organic matter to aquatic environments, Estuarine, Coastal and Shelf Science, 73, 65–72,
- https://doi.org/10.1016/j.ecss.2006.12.021, 2007.
- Phillips, D. L., Inger, R., Bearhop, S., Jackson, A. L., Moore, J. W., Parnell, A. C., Semmens,
- B. X., and Ward, E. J.: Best practices for use of stable isotope mixing models in food-web
- studies, Can. J. Zool., 92, 823–835, https://doi.org/10.1139/cjz-2014-0127, 2014.
- Polsenaere, P., Savoye, N., Etcheber, H., Canton, M., Poirier, D., Bouillon, S., and Abril, G.:
- Export and degassing of terrestrial carbon through watercourses draining a temperate
- podzolized catchment, Aquat Sci, 75, 299–319, https://doi.org/10.1007/s00027-012-0275-2,
- 2013.
- Pradhan, U. K., Wu, Y., Wang, X., Zhang, J., and Zhang, G.: Signals of typhoon induced
- hydrologic alteration in particulate organic matter from largest tropical river system of Hainan
- Island, South China Sea, Journal of Hydrology, 534, 553–566,
- https://doi.org/10.1016/j.jhydrol.2016.01.046, 2016.
- Regnier, P., Friedlingstein, P., Ciais, P., Mackenzie, F. T., Gruber, N., Janssens, I. A.,
- Laruelle, G. G., Lauerwald, R., Luyssaert, S., Andersson, A. J., Arndt, S., Arnosti, C., Borges,
- 1126 A. V., Dale, A. W., Gallego-Sala, A., Goddéris, Y., Goossens, N., Hartmann, J., Heinze, C.,
- Ilyina, T., Joos, F., LaRowe, D. E., Leifeld, J., Meysman, F. J. R., Munhoven, G., Raymond,
- P. A., Spahni, R., Suntharalingam, P., and Thullner, M.: Anthropogenic perturbation of the
- carbon fluxes from land to ocean, Nature Geosci, 6, 597–607,
- https://doi.org/10.1038/ngeo1830, 2013.
- Rousseeuw, P. J.: Silhouettes: A graphical aid to the interpretation and validation of cluster
- analysis, Journal of Computational and Applied Mathematics, 20, 53–65,
- https://doi.org/10.1016/0377-0427(87)90125-7, 1987.
- Sarma, V. V. S. S., Krishna, M. S., Prasad, V. R., Kumar, B. S. K., Naidu, S. A., Rao, G. D.,
- Viswanadham, R., Sridevi, T., Kumar, P. P., and Reddy, N. P. C.: Distribution and sources of
- particulate organic matter in the Indian monsoonal estuaries during monsoon, Journal of
- Geophysical Research: Biogeosciences, 119, 2095–2111,
- https://doi.org/10.1002/2014JG002721, 2014.
- Sato, T., Miyajima, T., Ogawa, H., Umezawa, Y., and Koike, I.: Temporal variability of
- stable carbon and nitrogen isotopic composition of size-fractionated particulate organic matter
- in the hypertrophic Sumida River Estuary of Tokyo Bay, Japan, Estuarine, Coastal and Shelf
- Science, 68, 245–258, https://doi.org/10.1016/j.ecss.2006.02.007, 2006.
- Savoye, N., Aminot, A., Tréguer, P., Fontugne, M., Naulet, N., and Kérouel, R.: Dynamics of
- particulate organic matter d15N and d13C during spring phytoplankton blooms in a
- macrotidal ecosystem (Bay of Seine, France), Marine Ecology Progress Series, 255, 27–41,
- https://doi.org/10.3354/meps255027, 2003.
- Savoye, N., David, V., Morisseau, F., Etcheber, H., Abril, G., Billy, I., Charlier, K., Oggian,
- G., Derriennic, H., and Sautour, B.: Origin and composition of particulate organic matter in a
- macrotidal turbid estuary: The Gironde Estuary, France, Estuarine, Coastal and Shelf Science,
- 108, 16–28, https://doi.org/10.1016/j.ecss.2011.12.005, 2012.

- Sigman, D. M., DiFiore, P. J., Hain, M. P., Deutsch, C., Wang, Y., Karl, D. M., Knapp, A. N.,
- Lehmann, M. F., and Pantoja, S.: The dual isotopes of deep nitrate as a constraint on the cycle
- and budget of oceanic fixed nitrogen, Deep Sea Research Part I: Oceanographic Research
- Papers, 56, 1419–1439, https://doi.org/10.1016/j.dsr.2009.04.007, 2009.
- Souissi, S., Yahia-Kéfi, O. D., and Yahia, M. N. D.: Spatial characterization of nutrient
- dynamics in the Bay of Tunis (south-western Mediterranean) using multivariate analyses:
- consequences for phyto- and zooplankton distribution, Journal of Plankton Research, 22,
- 2039–2059, https://doi.org/10.1093/plankt/22.11.2039, 2000.
- Sun, X., Fan, D., Cheng, P., Hu, L., Sun, X., Guo, Z., and Yang, Z.: Source, transport and fate
- of terrestrial organic carbon from Yangtze River during a large flood event: Insights from
- multiple-isotopes ( $\delta$ 13C,  $\delta$ 15N,  $\Delta$ 14C) and geochemical tracers, Geochimica et
- Cosmochimica Acta, 308, 217–236, https://doi.org/10/gn8qs5, 2021.
- Turner, R. E., Milan, C. S., Swenson, E. M., and Lee, J. M.: Peak chlorophyll a concentrations
- in the lower Mississippi River from 1997 to 2018, Limnology and Oceanography, n/a,
- https://doi.org/10/gpjm83, 2022.
- Veyssy, E., Etcheber, H., Lin, R. G., Buat-Menard, P., and Maneux, E.: Seasonal variation
- and origin of Particulate Organic Carbon in the lower Garonne River at La Reole
- (southwestern France), Hydrobiologia, 391, 113–126,
- https://doi.org/10.1023/A:1003520907962, 1998.
- Wang, X., Chen, Y., Yuan, Q., Xing, X., Hu, B., Gan, J., Zheng, Y., and Liu, Y.: Effect of
- river damming on nutrient transport and transformation and its countermeasures, Frontiers in
- Marine Science, 9, 2022.
- Wang, Y., Song, J., Duan, L., Yuan, H., Li, X., Li, N., Zhang, Q., Liu, J., and Ren, C.:
- Combining sterols with stable carbon isotope as indicators for assessing the organic matter
- sources and primary productivity evolution in the coastal areas of the East China Sea,
- Continental Shelf Research, 223, 104446, https://doi.org/10.1016/j.csr.2021.104446, 2021.
- Yaalon, D. H.: Soils in the Mediterranean region: what makes them different?, CATENA, 28,
- 157–169, https://doi.org/10.1016/S0341-8162(96)00035-5, 1997.
- Yan, X., Yang, J.-Y. T., Xu, M. N., Wang, H., Dai, M., and Kao, S.-J.: Nitrogen isotope
- constraint on the zonation of multiple transformations between dissolved and particulate
- organic nitrogen in the Changjiang plume, Science of The Total Environment, 818, 151678,
- https://doi.org/10.1016/j.scitotenv.2021.151678, 2022.
- Yu, F., Zong, Y., Lloyd, J. M., Huang, G., Leng, M. J., Kendrick, C., Lamb, A. L., and Yim,
- W. W.-S.: Bulk organic δ13C and C/N as indicators for sediment sources in the Pearl River
- delta and estuary, southern China, Estuarine, Coastal and Shelf Science, 87, 618–630,
- https://doi.org/10.1016/j.ecss.2010.02.018, 2010.
- Zhang, Y., Meng, X., Bai, Y., Wang, X., Xia, P., Yang, G., Zhu, Z., and Zhang, H.: Sources
- and features of particulate organic matter in tropical small mountainous rivers (SW China)
- under the effects of anthropogenic activities, Ecological Indicators, 125, 107471,
- https://doi.org/10.1016/j.ecolind.2021.107471, 2021.