# Peer review of "Toward a typology of river functioning: a"

_EGUsphere, 2025_

## Author Comment (AC1)

**Reviewer 1**

**Dear associate Editor of Biogeosciences journal,**

The work of Ferchiche et al deals with the construction of a typology of 23 French temperate rivers based on a set of national databases and previous works encompassing a wide range of parameters of POM (bulk and stable isotopic) able to determine the nature of POM (phytoplankton versus terrestrial OM, both labile and refractory and to a less extent, anthropogenic OM). Authors tried to link these parameters with many catchment properties belongs to anthropogenic, climatic and to a less extent geomorphological forcings; they naturally included the hydrological context (water, SPM discharges). In order to achieve their goal, they used Bayesian mixing model, seldomly used in our scientific community. In addition, POM signatures (sources) are not based on the sources or proxy (lake phytoplankton as a proxy for river ones) but used a methodology previously developed by the author able to track the change of POM parameters during the journey in river system or within its water column. For all these points, this work is original and deserves to be published in the journal.

However, this work suffers of many issues making its publication impossible in the present form, among them: (i) the lack of consideration of a significant source of POM in their work (litoshpheric), (ii) the sampling period of the rivers vs POM composition, (iii) the choice of the river systems (the over representation of the small rivers coming from the same area (SW France) associated to the size of the analysed watershed, and (iv) the choice of basin metrics and descriptors. Hereafter the detail of these 4 major issues.

We would like to thank Reviewer 1 for the comprehensive analysis of our work, pointing out the quality of the manuscript while identifying important ways to improve its form and content. We are glad to address all of Reviewer1's concerns.

**1-Lithospheric, (fossil or geological) POM source.** Authors consider 4 sources (3 majors): aquatic productivity (phytoplankton), labile and refractory terrestrial POM and anthropogenic POM. However, it is well known that rock-derived POM, having various labelling in many studies (lithospheric, geogenic, fossil etc) can be a substantial POM sources in rivers. As an example, authors should take a look in literature, especially for the well documented Rhone River which is also one of the studied rivers in this work. We need to wait the end of the ms (section 4.3, L578f) to see a possible lithospheric origin of refractory terrestrial POM. Authors should state at the beginning of this ms that terrestrial refractory could be lithospheric and mix with either deep soil horizons or surficial sediments stored in the catchments; and clearly rewrite their conclusion about the type II since almost 1/5 of the POC of the Rhone River is rock-derived. Additionally, the difference between soil and rocks refractory POM could be refined with C/N ratio where values for the fossil counterpart are normally excessively high (no more Nitrogen) compared to refractory POM soil-derived; as much as they can, authors should better explore the C/N ratio.

We agree with this statement. We will better introduce and integrate the bedrock potential origin of POM, and its influence on the refractory terrestrial source, given isotopic and elemental signatures. Effectively, some soil and rock-derived POM may be distinguished by their $\delta^{13}C$ or C/N values.

However, studies focused on petrogenic POM that measured directly this source showed a great variability of C/N ratio depending on the geology (at Taiwan island scale, C/N ratio ranges from

3 for sandstone to 17 for turbidites, and from 3.8 to 9.7 for rock-derived river sediment POM; Liu et al. 2017; Hilton et al. 2010).

In addition, the soil POM C/N ratio is correlated to the mineral-associated POM C/N ratio, and can also span a great range of values (Europe scale, 2 to 100 vs. 2 to 30, respectively; Cotrufo et al. 2019).

Finally, as rock-derived OM dominates the riverbed sediment OM in major rivers like the Amazon River (Sun et al. 2017) or the Rhône River (Harmelin-Vivien et al. 2010), and as riverbed sediment is subject to resuspension, we are keen to better address these geological pathways resulting in complex refractory terrestrial suspended POM sources.

**Cotrufo, M.F.**, Ranalli, M.G., Haddix, M.L. *et al.* Soil carbon storage informed by particulate and mineral-associated organic matter. *Nat. Geosci.* **12**, 989–994 (2019). https://doi.org/10.1038/s41561-019-0484-6
**Harmelin-Vivien, M.**, Dierking, J., Bănaru, D. *et al.* Seasonal variation in stable C and N isotope ratios of the Rhone River inputs to the Mediterranean Sea (2004–2005). *Biogeochemistry* **100**, 139–150 (2010). https://doi.org/10.1007/s10533-010-9411-z
**Robert G. Hilton**, Albert Galy, Niels Hovius, Ming-Jame Horng, Hongey Chen, The isotopic composition of particulate organic carbon in mountain rivers of Taiwan, Geochimica et Cosmochimica Acta, Volume 74, Issue 11, 2010, Pages 3164-3181, ISSN 0016-7037, https://doi.org/10.1016/j.gca.2010.03.004.
**Liu, K. K.**, and S. J. Kao, 2007: A three end-member mixing model based on isotopic composition and elemental ratio. Terr. Atmos. Ocean. Sci., 18, 1067-1075, doi: 10.3319/TAO.2007.18.5.1067(Oc).
**Sun, S.**, Schefuß, E., Mulitza, S., Chiessi, C. M., Sawakuchi, A. O., Zabel, M., Baker, P. A., Hefter, J., and Mollenhauer, G.: Origin and processing of terrestrial organic carbon in the Amazon system: lignin phenols in river, shelf, and fan sediments, Biogeosciences, 14, 2495–2512, https://doi.org/10.5194/bg-14-2495-2017, 2017.

**2-Sampling period (table 1).** I have a major concern about the sampling period: a significant number of rivers has only 1 year sampling. This is a serious bias as 1 year is far to be enough to capture the hydrological dynamics of the river. Indeed, since the control of the hydrology in this NW Europe is due to NAO index with a cycle of 4-8 years, for a sampling year, the hydrology could be lower or higher than it is expected. If the authors consider that the hydrology (water disharge) is a key point controlling the contribution of POM sources (their statement), then the average nature of POM is not well constraint...I think this a weakness point. I would suggest to not consider the rivers with only 1 year sampling period (at least) and authors should consider this bias in the discussion.

Indeed, our data acquisition strategy of 1-year river samplings does not allow for capturing the inter-annual hydrological dynamics of the rivers and is not representative of their overall hydrological dynamics. This was in fact not our aim. Our aim, however, was to describe 23 sets of POM sources dynamics and compare them with 23 sets of environmental forcings to establish a typology of POM dynamics. For this reason, we do not pretend to bind a typology to these particular 23 rivers, but rather bind POM sources dynamics with typical settings of temperate environmental forcings. This would imply that a river may belong to a type in a given year but to another type in another year.

For performing the fuzzy clustering in the manuscript (Fig. 5), we used, when a river was sampled during two or more years, only one of the sampled years. The idea was not to over-represent these rivers compared to one-year sampled rivers in the analysis. To check 1) if river type depends on sampling year and 2) for a possible bias between 1-year sampled rivers and multi-year sampled rivers due to over-representation of the latter, we performed another fuzzy clustering (same caption as Fig. 5), using all the sampling years. It results that the river typology is very similar to that of Fig. 5 and that each of the rivers that were sampled for more than a

year belongs to the same type, whatever the sampling year, except the Leyre River. The Leyre River switched from Type III in 2008 to Type I in 2009. Note that all the southwestern rivers belong to types I and III. The Leyre River is a southwestern river.

[Figure]

This will be better explained in the manuscript, and thanks to this comment, we will to better state the scope of our work.

**3-the choice of river systems .** Over the 23 studied rivers, a third (n=8) comes from the same close area, how can we make an appropriate typology with a such disequilibrium? Can we consider a bias with this over representation of these rivers characterizing the type I ? Why not decrease the number of these rivers and try to find another ones, outside the national territory?

We understand Reviewer 1's concern about the risk of representing the eight spatially closed little rivers. To test this potential bias, we performed again the fuzzy clustering, removing seven of these eight rivers (keeping only the Milieu River, representative of type I). It results that the typology is very similar to that of Fig. 5 and that Type I still stands with the Seudre and Milieu Rivers that still belong to this type. This indicates that this clustering is a robust analysis for performing such a typology because it is poorly dependent on sampling year and rivers, at least at the time and space scale of the study. This will be explained in section 2.6 of the revised manuscript.

[Figure]

**4-metrics .** The choice of the metrics is debatable (section 2.5, L289f). As an example, the size of the drainage basin for which the water discharge is closely dependent (mostly): since the authors highlight on the role of the water discharge in the POM composition, the size of the drainage basin should be a key parameter controlling this parameter and the proportions of land cover (the larger the basin area, the greater the proportion (in surface) of cultivated area, as an example).

Authors should focus on the common metrics used in literature which try explain the water discharge and SPM flux (which is also a key metric for terrestrial POM) and test these ones. As an example, why there is no relief-derived metrics as slope of the main river corridor, relief of the drainage basin? Why authors used geographic coordinates? In the end, authors should also explain why they selected their different parameters and why they pushed others aside (even if some of them disappear to limit the problem of auto-correlation), in a new sub-section 2. As an

example, the choice to keep the wetlands area, the wind, pH etc. need more explanations (even if most of these parameters are finally useful).

We thank Reviewer 1 for the wise tips on new parameters to take into account. We will acquire the necessary data and add these parameters to perform RDAs (slope and river connectivity).

Indeed, all metrics used to perform multivariate analyses need to be justified, here as translators of potential environmental forcings to the POM dynamics.

In the case of river-specific RDAs, environmental forcing dynamics may explain local seasonal dynamics, as seasonal meteorological or hydrological dynamics (e.g., high wind energy may be linked to resuspension processes). They also served to assess the robustness of the mixing model by linking known parameters to be influenced by the source (e.g., chlorophyll $a$ is a direct proxy of the living phytoplankton biomass, or pH may be indirectly lowered by living phytoplankton biomass).

In the case of the multi-rivers RDA, environmental forcings dynamics only explain the spatial dynamics between rivers, as the data input is only the annual mean. It permits to consider stable watershed forcings (on an annual scale), as soil properties and uses, or geographical coordinates, that may translate a latitudinal temperate climatic gradient (from degraded oceanic to Mediterranean climates).

We will consider this comment and improve the understandability of the different environmental drivers to POM dynamics.

Other minor comments, some of them belongs to these 4 major points.

L44f: I don't think this paragraph belongs to the abstract or if this paragraph is not at its place.

Will be considered.

L64f: it misses geological OM from erosion and weathering of sedimentary rocks.

Will be considered.

L70: compare to phytoplankton, yes, but mainly refractory is a bit exaggerated, this is a continuum of chemical composition leading with various labile and refractory POM.

Will be considered.

L78-81: authors should consider other drivers than river hydrodynamics as geomorphological, climatic, anthropogenic and even geological (lithologic) is their statement.

Will be considered.

L109f, 123: add the following idea: only in France with limited temperate river systems with an over-representation of small rivers in SW France.

Indeed, our work is restricted to Western Europe. It will be considered.

L156f: why the authors use wind data, it could be interested to use it for the objectives of this work, but authors should state why it is important.

See the 4-metrics issue answer.

L161f : I wonder if these spatial databases are at the same resolution, authors should at least give the resolution and explain the harmonization between databases to get an unique scale.

Technically, datasets for soil organic carbon data were retrieved as a raster, Corine Land Cover as a polygon map, and erosional soil data as a punctual map. Each dataset was cut according to each watershed, then averaged individually. This pattern ensures that each value is a real average of each watershed and that there is harmony between the resulting parameters. We will add this information to the manuscript.

L175 & 177f, 217: see general comment n°1

Will be considered.

L187: PN/SPM need reference

PN/SPM ratio was used to distinguish sources for rivers where consensual discriminators were not available (chlorophyll *a* and phaeopigments). Although it has not been documented as a phytoplankton descriptor, the PN/SPM ratio seemed appropriate for this source in our study.

L246, 288f: see general comment n°4

The pertinence of parameters used to qualify the isotopic variability of certain sources is discussed, as for the nitrates (that Reviewer 1 noted l.288f) at l.189-191 of the manuscript.

L317: give the name of the rivers

Will be considered.

L364: to be check with the format of the journal, but maybe no need to use mol/mol.

We understand the cumbersome nature of this notation. However, we would like to avoid any confusion about the mol/mol nature of the C/N ratio, while it cohabits with C/N ratios expressed in g/g in the literature (sometimes implicitly).

L365: which pigment concentration, phaeopigment or chlorophyl?

Chlorophyll *a*, phaeopigments, or the sum of both. Will be considered.

L375: values are a bit low for terrestrial plants, how you can explain that? (see Lamb et al., 2006, Earth Sci. Rev).

See the answer of the 1-Litospheric origin. We do not consider the refractory terrestrial POM originating from fresh terrestrial plants (or fresh litter; C/N ratio always > 12 mol/mol), but from eroded mineral-dominated soils or resuspended riverbed sediment, possibly associated with bacteria (same Fig. in Lamb et al. 2006). In that case, the low C/N ratio signatures of

refractory terrestrial POM are consistent with those of these sources (l. 501 to 508 of the manuscript).

L443 450f: terrestrial POM is commonly linked to SPM concentration, and due to surface processes, in turn SPM concentration is commonly linked to water discharge, but the direct link terrestrial POM concentration with water discharge is hard to believe. How explain the difference of labile (water discharge) and refractory (SPM) links since most of these materials comes from soil erosion.

We agree with Reviewer 1 that high SPM is usually driven by flood, so high river flows. The slight decoupling shown in the redundancy analyses has multiple causes.

In the Milieu River, since there is a dominance of terrestrial matter (so a weak source dynamics), SPM is strongly correlated with the POC dynamics, so both POC sources, while river flow induced more POC of terrestrial origin, where the minimum of phytoplankton proportion was found.

In the Charente and Rance Rivers, if river flow is correlated with more labile terrestrial POM than SPM, the latter is also linked with phytoplankton bloom, responsible for bulk POC and SPM peaks. Moreover, high phytoplankton proportions may also be linked with low river flows.

In the Hérault River, even if they are closer to one terrestrial source, both river flows and SPM are correlated with both terrestrial sources. This slight decoupling may come from processes of resuspension that can be triggered by other forcings than high river flows (as wind, poorly shown here).

We will consider this commentary to improve the RDAs discussion.

L469f, 564f: how can you explain that? the higher the temperature, the higher the proportion of refractory terrestrial OM? In addition, most of the time, erosion in this area (with the exception of heavy rainfall) occurs in autumn, winter and spring times, where the temperature is pretty low…

The multi-site RDA is set as an average year, so without any temporal information (see 4-metrics answer). Here, the rivers with the highest average temperature are the Mediterranean ones, logically, the only ones where we considered refractory terrestrial POM. Since this bias is understood, no real environmental information can be extracted from this correlation.

L543f: why there is no discussion about refractory terrestrial OM and their relation with metrics? this was however in the result section - even if it is hard to believe the link with temperature. Also: authors used a lot of parameters (quite disputable, especially geomorphic parameters but also climatic wind)), but there is no words about the absence of discussion (even add sentence about the bad relations, as seen in figure 7).

We agree. A discussion will be added about the environmental drivers to the refractory terrestrial dynamics across rivers, given the new RDA performed with more pertinent parameters (slope, connectivity).

L577f: see general comment n°1, I quite disagree, since: there is no consideration for geological OM, no consideration of sedimentary cascade which increase terrestrial POM processes in river

corridors - the link the authors highlighted is mainly due to their choice they made. river connectivity for example should be interesting to test as the slope of the main river corridor. such parameters can be easily calculated, I suggest to test other significant parameters rather to be speculative with sediment resuspension (even this could have a link).

In addition to the precedent answer, we assure that we will revise our hypothesis depending on the new results, better integrating watershed processes.

L632f: add with our definition of the labile or refractory character at the beginning of this sentence. Also considering the fig.6, labile terrestrial POM is also linked to SPM. In the end, I suggest to use a correlation matrix and other statistical tests.

Indeed, correlations between the environmental parameters are not treated here. We will consider adding statistical univariate correlations when necessary.

L642: in that sense, terrestrial POM should be linked with SPM too? why this is not the case.?

After checking, SPM was removed from the multi-site RDA for its weakness in the variance representation (short arrow). It seems that rivers with an average phytoplankton dominance induced as high SPM concentrations as rivers with an average terrestrial dominance.

Figure A4-A5: Loire is missing

You are right. Since the Loire River sources proportions and RDA had already been published (Ferchiche et al. 2024), we decided not to duplicate these specific results in this manuscript.

Ferchiche F, Liénart C, Charlier K, Coynel A, Gorse-Labadie L and Savoye N (2024) Quantifying particulate organic matter: source composition and fluxes at the river-estuary interface. *Front. Freshw. Sci.* 2:1437431. doi: 10.3389/ffwsc.2024.1437431

Call of references in the text: follow the order of publication year according the journal (younger to older or vice versa).

Will be considered.

---

## Author Comment (AC2)

**Reviewer 2**

Florian et al. quantified and examined relationships between POM composition and environmental forcings in 23 rivers in France using Bayesian mixing models associated with statistical multivariate analyses. Determining the POM composition and its controlling factors in multiple rivers provides valuable insights for better understanding the carbon cycling along the Land-Ocean Aquatic Continuum. However, it still needs a number of improvements for novelty and discussion. Following is my major concerns and specific comments on the manuscript.

We would like to thank Reviewer 2 for providing a thorough analysis of our work, highlighting the manuscript's quality while offering important suggestions to improve its structure and substance. We are glad to address all of Reviewer 2's concerns.

Major                                                                                    concerns

(1)   In this study, the authors analyzed POM composition in 23 rivers in France. However, these rivers are geographically clustered within a similar environmental setting, exhibiting closely comparable temperatures and similar POC composition (C/N and δ13C). These similar features may limit the broader implications of the findings. Thus, I suggest that the authors reconsider and summarize the novelty of the work.

Indeed, the 23 rivers we chose to gather into this study are clustered in the same temperate climate. We will improve the manuscript to better explain the climatic range of our study in a somewhat restrained geographic area and the broad implications of our approach, at least from a technical point of view.

(2)   It appears that petrogenic OC was neglected in the mixing model. Petrogenic OC, derived from sedimentary rocks, represents a fossil-derived OC component and constitutes a significant fraction of riverine POM. Furthermore, petrogenic OC is generally characterized by higher $\delta^{13}C$ values (typically ranging from -20‰ to -23‰) compared to OC derived from soil or plant. Thus, lack of petrogenic OC could introduce significant uncertainties into the model results.

Reviewer 2 raises an important point: we did not consider any unique petrogenic source in our mixing models. However, we considered that the refractory terrestrial POM sources we distinguished from the bulk POM can be of petrogenic origin.

Indeed, as raised by Reviewer 2, petrogenic OM, according to the geology, can be defined by a $\delta^{13}C$ between -20 and -26 ‰ (Hilton et al. 2010). Also distinguished by low C/N ratios for a terrestrial source, they can be easily integrated into isotopic mixing models.

For the 23 rivers, rare were the bulk suspended POM values that were higher than -26 ‰ (only a few ones for the Têt and the Rhône Rivers). Thankfully, our method permitted us to discriminate a refractory terrestrial POM source for these two rivers. Since this, we can consider that for the Têt and Rhône rivers, this source is at least influenced by petrogenic POM.

We will better describe the possible petrogenic origin of this source in the manuscripts with the help of this commentary. Also, see our response to Reviewer 1 (first major issue).

**Robert G. Hilton**, Albert Galy, Niels Hovius, Ming-Jame Horng, Hongey Chen, The isotopic composition of particulate organic carbon in mountain rivers of Taiwan, Geochimica et Cosmochimica Acta, Volume 74, Issue 11, 2010, Pages 3164-3181, ISSN 0016-7037, https://doi.org/10.1016/j.gca.2010.03.004.

(3)    Determining a typology of rivers according to their POM composition and dynamics is a key point of this study. However, the analysis is currently focused on a specific regional context. To enhance the broader significance of this work, I suggest that the authors address the potential global transferability or applicability of their proposed typologies.

Absolutely, the actual work is focused on temperate Rivers in Western Europe. Nevertheless, the approach that we used in the manuscript can be applicable to other climates and broadened to a greater range of environmental conditions, including different climatic settings (polar to tropical) and geographical locations (catchment properties) (l.655-658 of the manuscript). This will be better detailed in section 5 and in the abstract.

Specific                                                                                                    comments

1.    The abstract is a little bit long and contains a lot of redundant parts. I suggest that the authors reorganize the abstract. For example, the content in line 44-46 is redundant as before.

Will be considered.

2.    Line 102-103, Actually, a quick literature review showed that many previous works already using mixing models to quantify POM composition in river systems. Consequently, characterizing such analyses as "scarce" appears inconsistent with established research. I suggest reconsider the novelty of this work around more specific research gaps or methodological advancements.

Reviewer 2 suggests that POM composition mixing models are common in rivers.

Indeed, since the 1990s, a few dozen articles have been published, using mixing models to partially or completely quantify POM sources' proportions in freshwater rivers. However, as for Hilton et al. (2010) previously cited, all these articles (except those of l.102-103 of the manuscript) performed outdated deterministic linear mixing models. These models only measure the distances between end-member signatures and the mixture value to give the proportion of the distances. A part of this literature focuses only on a fraction of the POM, as it only accounts for terrestrial origins, or fails to correctly distinguish phytoplankton source (Hilton et al. 2010, Dalu et al. 2016, Lu et al. 2016).

Bayesian (probabilistic) mixing models published in the 2010s until today calculate proportions using isotopic and elemental signatures and their uncertainties, and introduce priors, allowing to give an interval of confidence for each mixing model, and uncertainties to the source's proportions. The credibility of the whole model is then measurable, contrary to frequentist models.

For this reason, particular attention should be paid to the methods used to quantify sources of POM. We would like to reassure Reviewer 2 that the use of the Bayesian mixing model is rare (as pointed out by Reviewer 1), and is unique by the integration of our methodology of source

discrimination and taking into account isotopic variability by source is a methodological advancement for riverine biogeochemistry.

**Robert G. Hilton**, Albert Galy, Niels Hovius, Ming-Jame Horng, Hongey Chen, The isotopic composition of particulate organic carbon in mountain rivers of Taiwan, Geochimica et Cosmochimica Acta, Volume 74, Issue 11, 2010, Pages 3164-3181, ISSN 0016-7037, https://doi.org/10.1016/j.gca.2010.03.004.
**Dalu, T.**, Richoux, N.B. & Froneman, P.W. Nature and source of suspended particulate matter and detritus along an austral temperate river–estuary continuum, assessed using stable isotope analysis. *Hydrobiologia* **767**, 95–110 (2016).                                                                  https://doi.org/10.1007/s10750-015-2480-1
**Lu Lu**, Hongguang Cheng, Xiao Pu, Jiantong Wang, Qianding Cheng, Xuelian Liu, Identifying organic matter sources using isotopic ratios in a watershed impacted by intensive agricultural activities in Northeast China,Agriculture, Ecosystems & Environment,Volume 222, 2016, Pages 48-59, ISSN 0167-8809, https://doi.org/10.1016/j.agee.2015.12.033.

3.    Line 503-508 and line 561-564. Some simple comparisons were made, but the authors did not explain whether these rivers have similar of environmental forcings and/or climate characteristics.

Reviewer 2 raised a good point. Features common to these rivers will be added.

4.    Line 594. The section 4.4 primarily summarizes findings rather than providing critical interpretation. I suggest that this section should be condensed and incorporated into Section 5.

Will be considered.

---

## Author Response (AR2)

Dear Florian Ferchiche,

Thank you for submitting the revised version of your manuscript, which was re-evaluated by the original referees.

Both reviewers acknowledged the substantial effort you have invested and the significant improvements you made. One of the reviewers, however, provided additional suggestions for further strengthening the manuscript.

I agree with these comments and encourage you to incorporate the suggested changes in your next revision.

Best regards,
Yuan Shen
Associate Editor

We want to thank editor Yuan Shen for moving the manuscript forward to the minor revisions stage, in light of our improvements. You will read below how we took into account the suggestions of R1 to improve the manuscript even more.

Comments:
It remains few concerns before a final acceptation. This new version was greatly improved, I thank the authors for that. However, I have comments which can make the ms less confuse.

We want to thank R1 for appreciating our modifications, taking the time to review them, and for giving us new and precious insights to improve the manuscript.

My concern still focuses on: sampling period, metrics and title:
1-Previous point 2: sampling period
I thank the authors for their thoroughly reply and changes which help the understanding of this part. However, this is not clearly explicated at the end of the introduction section, notably for the goal n°4: "determining a typology of river dynamics according to their POM composition and dynamics" (L110-111). Notably the fact that, for a river and depending of the year, the considered river may change the type.

Thanks to this comment, we improved the explanation around the scope of the river dynamics typology. Unfortunately, we did not manage to add this information within the sentence dedicated to the aims (end of section Introduction) but we add this information on l.333-334 in section 2.6, where we found it the most relevant.

2- title: I suggest to change the title, by introducing French rivers since only French rivers are tested.
R1 is right, writing that the 23 studied rivers all have their mouths in France. However, we do not consider that our study is dedicated to French rivers but rather that we considered these 23 French rivers as good representatives of temperate rivers. Indeed, they encompass a large gradient of temperate conditions, from continental to oceanic and Mediterranean climate, size and basin characteristics. Thus, we think that adding 'French' in the title would narrow the scope of the study that also aims to set example for studies accounting other climates.

3- Previous point 4: metrics
Sections 2.5 and 4.2 have been significantly improved in this revised version. However, there is a lack of details making the understanding a bit confuse and unclear - among them:
L313: river flow is not only climatic dependent but also geomorphological dependent. Latitude and longitude are not climatic dependent

You are right, river flow has been added as a geomorphological dependent l.313. Latitude and longitude are not climate-dependent, but latitude and longitude may be correlated to spatial climatic gradients.

L315: change hydromorphology by geomorphology, precise which slopes (channel, catchment ?)

Done and specified in manuscript l.313 and 509.

L316: what is artificial land use: urban others?

Yes, artificial coverage includes the urban areas, but also industrial and commercial areas. We use now precisely the official nomenclature of Corine Land Cover (first level) to avoid any misunderstanding.

L317: add water to useful reserves + soil type: there is a mix between soil classification WRB (podzol) and others (brown etc)

Done. The four major types of soil according to the WRB classification have been better named and are gathered accordingly:

- Acidic = Podzol, podzoluviol, regosol, arenosol and ranker;
- Brown = Cambisol, luvisol, phaeozem, chernozems, greyzem, vertisol, kastanozema and andosol;
- Hydromorphic = gleysol and all gleyic soils (e.g. gleyic cambisol);
- Organic and others = Histosol, rendzina, solonetz, planosol, xerosol, solonchalk, glacier, plaggensol and rock outcrop.

L318: geological type: there is a mix between surficial formations, lithology, granulometry, I suggest to reorganise this part. Loamy is rather used for soil than rocks.

You are right, the dataset providers described it more as bedrock origin or parent material types than geological type, names, and presentation have been modified accordingly.

Table A2: add the unit of parameters – I also suggest to add a reference for each of these parameters and to give a better description of these parameters in the table that deserved to be in the text and not in annexe (for example what is the useful reserve of water). This section 2.5 needs to be improved since these parameters will be used for the classification river dynamics.

Units added to table A2 and figure A6.

We understand that the nature of the "useful water reserve" parameter is not intuitive. We defined it specifically in a footnote in l.316.

We understand that introducing each parameter by a description and referenced links with biogeochemical processes would be complete. We agree but think that this will too much increase the length of the section and finally the manuscript. Nonetheless, these parameters are categorized depending on the processes they are involved in (Section 2.5, l. 305-322) and are discussed when correlated with POM sources dynamics in Section 4.2.

L606-607: why the refractory OM are negatively correlated to useful reserve of water in soils? authors should give more information, this is not intuitive.

Added l.612